# Sex difference in BAT thermogenesis depends on PGC-1α–mediated phospholipid synthesis in mice

Akira Takeuchi[1,4], Kazutaka Tsujimoto [1,4]✉, Jun Aoki[1,4], Kenji Ikeda [1], Nozomu Kono [2], Kuniyuki Kano [2], Yoshihiro Niitsu[1], Masato Horino[1], Kazunari Hara [1], Rei Okazaki[1], Ryo Kaneda[1], Masanori Murakami[1], Kumiko Shiba[1,3], Chikara Komiya[1], Junken Aoki [2] & Tetsuya Yamada[1]✉

Brown adipose tissue (BAT), a thermogenic tissue that plays an important role in systemic energy expenditure, has histological and functional sex differences. BAT thermogenic activity is higher in female mice than in male mice. However, the molecular mechanism underlying this functional sex difference has not been fully elucidated. Herein, we demonstrate the role and mechanism of PGC-1α in this sex difference. Inducible adipocyte-specific PGC-1α knockout (KO) mice display mitochondrial morphological defects and decreased BAT thermogenesis only in females. Expression of carbohydrate response-element binding protein beta (*Chrebpβ*) and its downstream de novo lipogenesis (DNL)-related genes are both reduced only in female KO mice. BAT-specific knock-down of ChREBPβ displays decreased DNL-related gene expression and mitochondrial morphological defects followed by reduced BAT thermogenesis in female wild-type mice. Lipidomics reveals that, PGC-1α increases ether-linked phosphatidylethanolamine (PE) and cardiolipin$(18:2)_4$ levels through Chrebpβ-dependent and -independent mechanisms in female BAT. Furthermore, PGC-1α enhances the sensitivity of female BAT estrogen signaling, thereby increasing *Chrebpβ* and its downstream DNL-related gene expression. These findings demonstrate that PGC-1α–mediated phospholipid synthesis plays a pivotal role in BAT thermogenesis in a sex-dependent manner.

Obesity is a major risk factor for type 2 diabetes mellitus, various metabolic diseases, and cardiovascular diseases, and its growing prevalence is a serious public health crisis[1]. Conversely, while the frequency of obesity in women is reported to be comparable to that in men or slightly higher[1,2], the prevalence of diabetes or cardiovascular disease, especially in premenopausal women, is clearly lower than that in men of the same ages[2,3]. Although there have been various reports on the "metabolic advantage of women" from the viewpoints of sex

hormones, chromosomes, and lifestyle[4], the underlying molecular mechanism has not yet been fully elucidated.

Brown adipose tissue (BAT) has the unique ability to catabolize energy substrates and release them as thermal energy, and numerous reports have established the importance of BAT in human energy metabolism[5]. In recent years, it has also been reported that the presence of BAT is inversely correlated with the risk of diabetes and cardiovascular disease[6]. Therefore, BAT is expected to be a promising

[1]Department of Molecular Endocrinology and Metabolism, Graduate School of Medical and Dental Sciences, Institute of Science Tokyo, Tokyo, Japan. [2]Laboratory of Health Chemistry, Graduate School of Pharmaceutical Sciences, University of Tokyo, Tokyo, Japan. [3]The Center for Personalized Medicine for Healthy Aging, Institute of Science Tokyo, Tokyo, Japan. [4]These authors contributed equally: Akira Takeuchi, Kazutaka Tsujimoto, Jun Aoki. ✉e-mail: ktsujimoto.mem@tmd.ac.jp; tyamada.mem@tmd.ac.jp

target for the treatment of these diseases. Notably, it has been reported that BAT in females surpasses that in males in terms of prevalence, quantity, and metabolic activity, exhibiting a higher level in each of these aspects[6]. Therefore, BAT is considered a potential mechanism contributing to the metabolic advantage observed in women.

Studies in rodents have already revealed that BAT has the potential to reduce obesity and impaired glucose tolerance by increasing energy expenditure[7]. Moreover, several reports have shown histological and functional sex differences in rodent BAT, including that female rat BAT has larger and more densely populated mitochondria than males, that female BAT initiates lipolysis with weaker β3-adrenergic stimulation[8], and that uncoupling protein-1 (UCP1), which is responsible for thermogenesis in BAT, is more highly expressed in female[9]. These findings indicate that female BAT has higher metabolic activity than male BAT. However, the molecular mechanism underlying these sex differences in the function of BAT has not been fully elucidated.

Peroxisome proliferator-activated receptor-γ coactivator 1α (PGC-1α) is a transcriptional coactivator that acts as a master regulator of mitochondrial metabolism and is responsible for the transcription of genes involved in the mitochondrial electron transport chain, fatty acid oxidation, and oxidative stress management in response to cellular energy demand[10]. Although PGC-1α is regarded as a key regulator of thermogenesis in BAT, a report on adipocyte-specific PGC-1α knockout male mice[11] showed only minor changes in BAT gene expression and histology, and the prominent metabolic phenotype observed was impaired glucose tolerance rather than cold intolerance[11]. Therefore, the in vivo role of PGC-1α in BAT has not been fully clarified.

In this study, we identify a sex difference in PGC-1α function in BAT and further show that PGC-1α in the BAT of female mice serves a unique role in thermogenesis regulation, distinct from its function in males, and plays a pivotal role in regulating systemic energy expenditure.

## Results

### PGC-1α deletion in BAT suppresses thermogenesis and impairs acute cold tolerance only in female mice

We first examined PGC-1α gene and protein expression in BAT of male and female mice. Female mice BAT expressed significantly higher PGC-1α compared to male mice, whereas there was no sex difference in white adipose tissue (WAT), and its expression was much lower than that in BAT at room temperature (Fig. 1a, b). Females also exhibited higher *Ucp1* expression than males (Fig. S1a). Sex differences in *Pgc1a* expression in BAT were attenuated under thermoneutral conditions and enhanced under cold exposure (Fig. 1c). In addition, the adenylate cyclase 3 (*Adcy3*) gene, which has been reported to be a marker gene of brown adipocytes[12] and cause severe obesity by its loss-of-function mutation in human[13,14], exhibited higher expression in female mice than in male mice (Fig. S1b) and showed a strong positive correlation with *Pgc1a* expression in both sexes (Fig. S1c). Moreover, the cAMP level in BAT upon norepinephrine (NE) administration was significantly higher in females than in males (Fig. S1d). These results suggest that equivalent adrenergic stimulation results in more robust intracellular signaling in female BAT, which may contribute to the sex difference in *Pgc1a* gene expression.

Next, to explore the biological significance of *Pgc1a* highly expressed in female BAT, we generated acquired adipocyte-specific PGC-1α knockout (KO) mice using tamoxifen (TMX)-inducible *Adipoq*-Cre-ERT2 mice, which express Cre recombinase under the regulatory elements of *Adipoq* (referred to as Male control, Male KO, Female Control, and Female KO, respectively). Considering the possible effect of TMX on sex differences, the following analyses were performed at least 8 weeks after administration (Fig. S1e). We confirmed that *Pgc1a* was deleted in the BAT, while no changes were observed in *Pgc1b* expression (Fig. S1f, g).

Using sexually mature male and female PGC-1α KO mice, rectal temperature was measured under acute cold exposure, along with controls. While the rectal temperature of Male KO mice did not show any significant difference compared with that of Male Control mice, Female KO mice exhibited a markedly lower rectal temperature compared with that of Female Control mice (Fig. 1d). Similar results were obtained from the thermographic measurement of the interscapular surface temperature, an indicator of BAT-derived thermogenesis (Fig. 1e, f). We additionally measured the enhancement of oxygen consumption (VO$_2$) after NE administration, which mainly reflects BAT function. Male KO mice showed no significant difference in VO$_2$ compared with Male Control mice, whereas Female KO mice showed a marked decrease in VO$_2$ compared with Female Control mice (Fig. 1g). On the other hand, the glucose tolerance test showed that glucose tolerance was markedly reduced in Male KO mice (Fig. S1h), which is consistent with the result of male congenital adipocyte-specific PGC-1α knockout mice reported in a previous study[11]. These results suggest that PGC-1α in female BAT plays a crucial role in thermogenesis upon adrenergic stimulation, possibly contributing to the cold tolerance of female mice.

### PGC-1α deletion abolishes the augmented mitochondrial membrane structure specific to female BAT

Since PGC-1α facilitates mitochondrial biogenesis[10], we next examined the mitochondrial morphology in BAT of Control and KO mice of both sexes. We found that Female Control mice had larger mitochondria and denser cristae compared with Male Control mice by electron microscopy imaging of BAT (Fig. 2a, b). Quantitative analysis showed a significantly longer total cristae length per mitochondrion in Female Control mice than in Male Control mice (Fig. 2c). However, only Female KO mice exhibited a decrease in mitochondrial size and disruption or loss of cristae, resulting in a significant reduction in the total cristae length per mitochondrion. (Fig. 2a–c).

Assuming that PGC-1α deletion is involved in the observed mitochondrial membrane structure degradation in BAT of Female KO mice, we then performed protein expression analysis of mitochondrial electron transport chain complexes in BAT of Control and KO mice of both sexes. We observed higher protein expression of electron transport chain complexes in Female Control mice than in Male Control mice, particularly that of Complex I and Complex IV. However, the expression of these complexes was reduced due to PGC-1α deletion only in female mice (Fig. 2d). Nuclear respiratory factor 1 (*Nrf1*) and Transcription factor A (*Tfam*), which have been reported to play an important role in mitochondrial regulation by PGC-1α[15,16], as well as the genes responsible for mitochondrial fusion/fission (Mitofusin-1; *Mfn1*, Mitofusin-2; *Mfn2*, Optic atrophy 1; *Opa1* and Dynamin-related protein 1; *Drp1*), were expressed significantly higher in females than in males, but were unchanged or mildly reduced in females due to PGC-1α deletion (Fig. S2). These results prompted us to the hypothesis that PGC-1α in female BAT may regulate mitochondrial morphology and function in a manner other than canonical pathways, such as transcriptional regulation of *Nrf1*/*Tfam* or mitochondrial fusion/fission-related genes.

### PGC-1α deletion decreases chromatin accessibility and transcription of *Chrebpβ* only in female BAT

To investigate the mechanism by which PGC-1α in female BAT contributes to enhanced thermogenesis and augmented mitochondrial membrane structure, RNA-seq analysis was performed on BAT from Control and KO mice of both sexes. The expression of 182 genes was downregulated in Male KO mice compared with Male Control mice, and 240 genes were downregulated in Female KO mice compared with Female Control mice. Only 47 genes were downregulated in both sexes (Fig. 3a). These results suggest that the genes transcriptionally regulated by PGC-1α are widely different between male and female mice. We

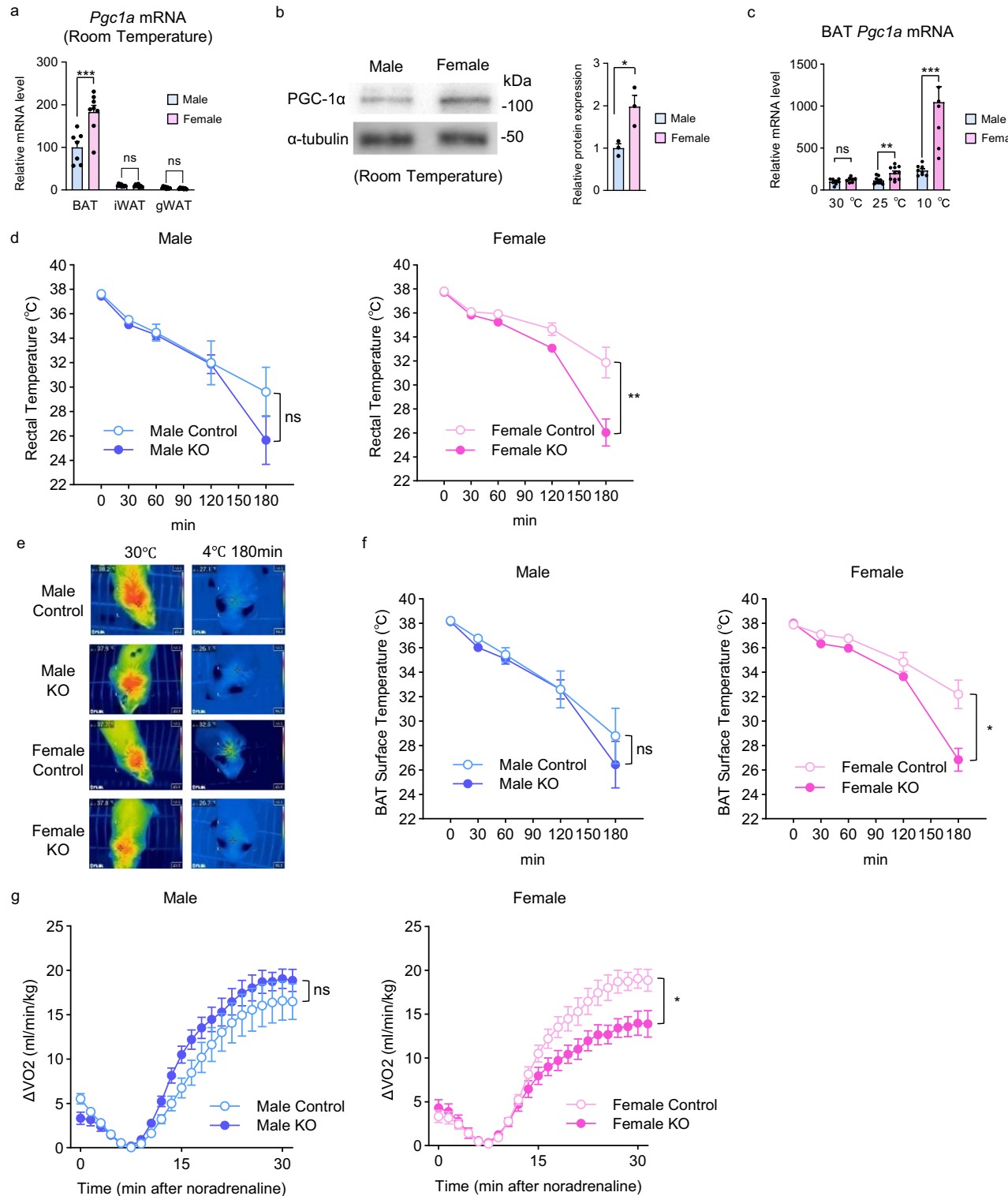

**Fig. 1 | PGC-1α is highly expressed in BAT and is essential for maintaining cold tolerance in female mice. a** *Pgc1a* mRNA levels in BAT ($n = 7$ Male and $n = 8$ Female, $p = 0.0001$), inguinal WAT (iWAT) ($n = 8$ Male and Female), and gonadal WAT (gWAT) ($n = 8$ Male and $n = 7$ Female). **b** Western blots for PGC-1α proteins in BAT (left) and protein levels normalized to α-tubulin (right). $n = 3$ per group. $p = 0.0124$. **c** *Pgc1a* mRNA levels in BAT at 30 °C ($n = 8$ Male and $n = 7$ Female), 25 °C ($n = 10$ Male and Female, $p = 0.0025$), and 10 °C ($n = 8$ Male and Female, $p = 0.0003$). **d** Rectal temperature in Male Control ($n = 7$), Male KO ($n = 8$), Female Control ($n = 5$), and Female KO ($n = 8$) mice under acute cold exposure. $p = 0.4146$ (Male Control vs. Male KO), $p = 0.0080$ (Female Control vs. Female KO). Representative thermal

images of interscapular surface (**e**) and interscapular surface temperature (**f**) in Male Control ($n = 7$), Male KO ($n = 8$), Female Control ($n = 5$), and Female KO ($n = 8$) mice under acute cold exposure. $p = 0.5386$ (Male Control vs. Male KO), $p = 0.0103$ (Female Control vs. Female KO). **g** Oxygen consumption (VO₂) recordings in response to NE in Male Control ($n = 8$), Male KO ($n = 8$), Female Control ($n = 7$), and Female KO ($n = 8$) mice. $p = 0.2275$ (Male Control vs. Male KO), $p = 0.0214$ (Female Control vs. Female KO). Data are expressed as the mean ± SEM. Data were analyzed using unpaired two-sided t-test (**a**–**c**) and two-way repeated measures ANOVA (**d**, **f**, **g**). Significance is indicated (*$p < 0.05$; **$p < 0.01$; ***$p < 0.001$). ns not significant. Source data are provided as a Source Data file.

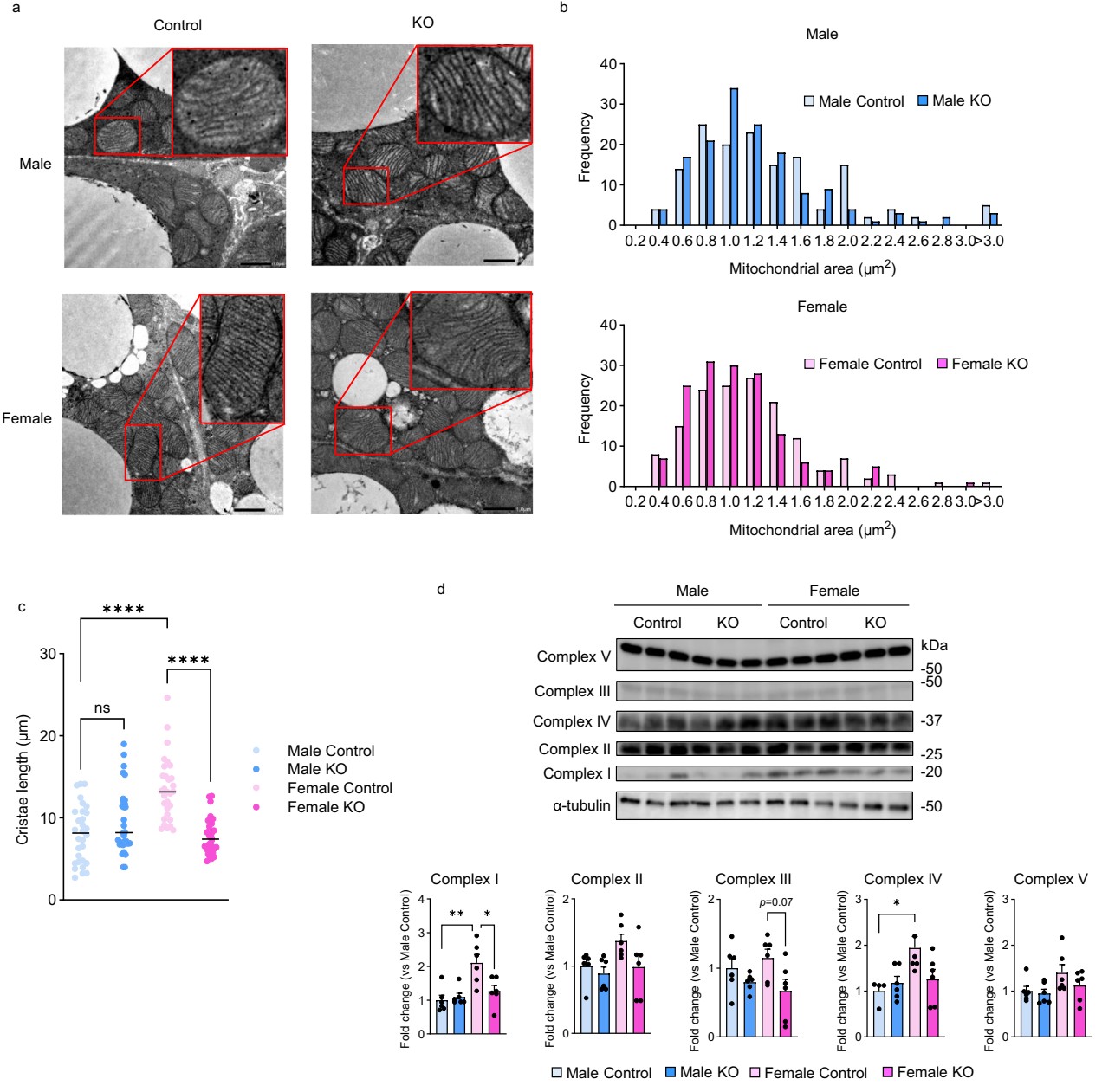

**Fig. 2 | PGC-1α modulates mitochondrial membrane structure in female BAT cells. a** Representative electron micrographs of mitochondria from the BAT of Control and KO mice. The areas outlined in red boxes indicate enlarged images of mitochondria. Scale bars = 1 μm (*n* = 3 biologically independent experiments). **b** Histograms showing the distribution frequency (%) of mitochondrial section areas (0–3 μm²). **c** Total cristae length per mitochondrion. *n* = 30 per group. *p* = 0.2604 (Male Control vs. Male KO), *p* < 0.0001 (Male Control vs. Female Control), *p* < 0.0001 (Female Control vs. Female KO). **d** Representative blots for electron transport chain complexes (upper panel) and protein levels normalized to α-tubulin (lower panel); Complex I (*n* = 6 per group), Complex II (*n* = 6 Male Control, *n* = 6 Male KO, *n* = 5 Female Control, and *n* = 6 Female KO), Complex III (*n* = 6 per group), Complex IV (*n* = 4 Male Control, *n* = 6 Male KO, *n* = 5 Female Control, and *n* = 6 Female KO) and Complex V (*n* = 6 per group). *p* = 0.0014 (Complex I, Male Control vs. Female Control), *p* = 0.0157 (Complex I, Female Control vs. Female KO), *p* = 0.0275 (Complex IV, Male Control vs. Female Control). Data are expressed as the mean ± SEM. Data were analyzed by one-way ANOVA with Tukey's post hoc test (**c**, **d**). Significance is indicated (**p* < 0.05; ***p* < 0.01; *****p* < 0.0001). ns not significant. Source data are provided as a Source Data file.

next performed gene ontology analysis of genes downregulated only in female mice due to PGC-1α deletion, which revealed enrichment in central carbon metabolism-related pathways such as "NADH regeneration, canonical glycolysis, and fatty acid derivative biosynthetic process" (Fig. 3b). In particular, the expression of genes related to de novo lipogenesis (DNL) and carbohydrate response-element binding protein beta (*Chrebpβ*), a transcriptional regulator of DNL-related genes[17], was markedly downregulated only in Female KO mice (Fig. 3c). In contrast, the expression of *Chrebpα*, a splice variant of *Chrebpβ*, did not differ between control and KO mice in either sex (Fig. S3a). The

total protein levels of ChREBP were higher in Female Control mice than in Male Control mice, and PGC-1α KO significantly reduced ChREBP protein levels only in female mice (Fig. S3b). Notably, the expression of *Fgf21*, a molecule which enhances BAT thermogenesis[18] and is implicated in the sexual dimorphism of energy metabolism[19], did not differ between Control and KO mice of either sex (Fig. S3c). These results suggest that PGC-1α in female BAT is responsible for the transcriptional regulation of *Chrebpβ* and genes related to DNL.

Although the direct regulation of *Chrebpβ* transcription by PGC-1α remains unreported, PGC-1α can influence the transcription of target

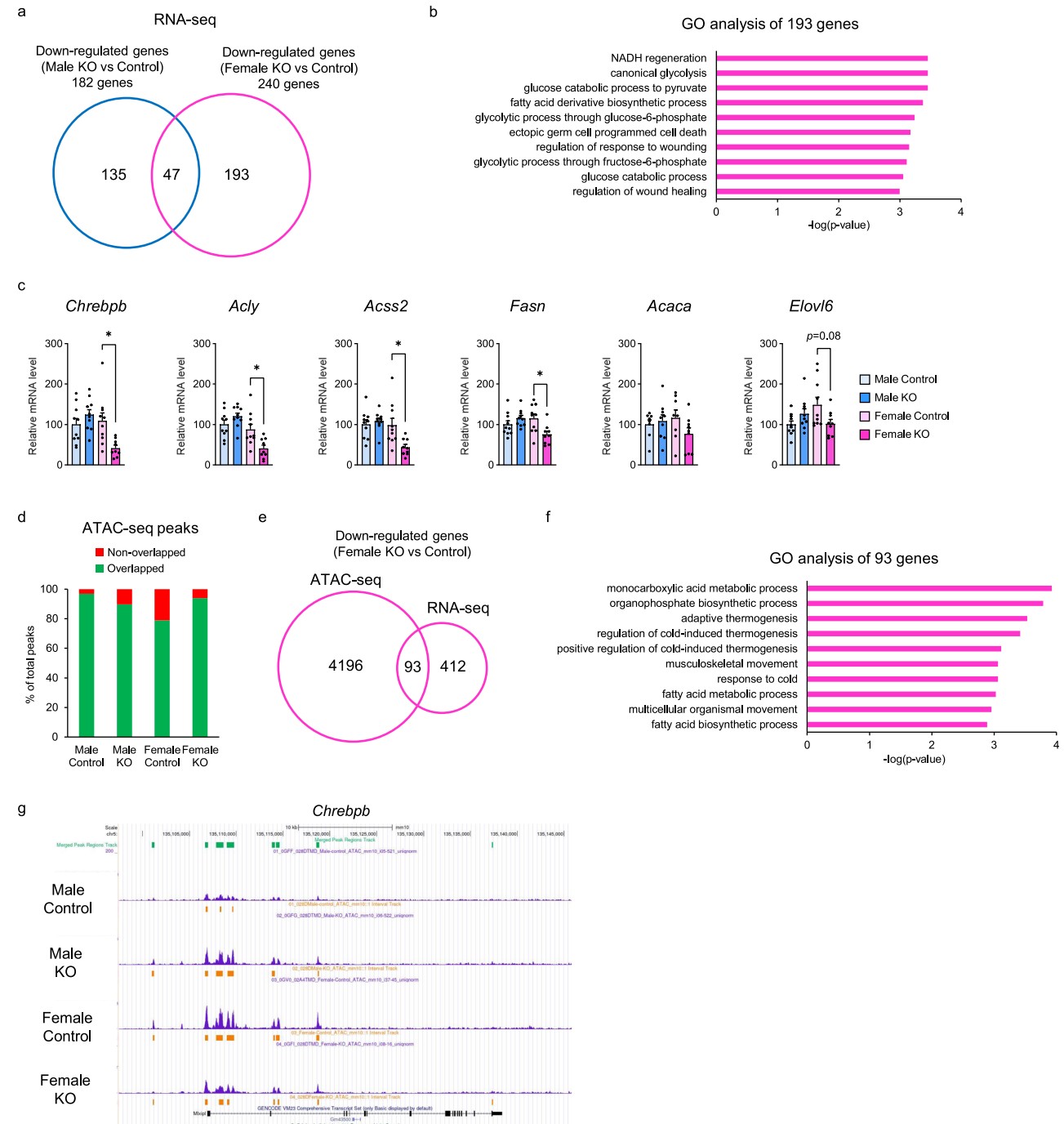

**Fig. 3 | PGC-1α regulates *Chrebpβ* gene expression by modulating chromatin accessibility in female BAT cells. a** Venn diagram showing the number of down-regulated genes in Male and Female KO mice compared with Controls. **b** Gene Ontology (GO) analysis of genes downregulated only in Female KO mice. The top 10 enriched terms showing Metascape-generated enrichment p-values using cumulative hypergeometric distributions. **c** Gene expression of *Chrebpβ* ($n = 10$ per group), *Acly* ($n = 10$ per group), *Acss2* ($n = 10$ per group), *Fasn* ($n = 10$ per group), *Acaca* ($n = 8$ Male Control, $n = 9$ Male KO, $n = 8$ Female Control, and $n = 10$ Female KO) and *Elovl6* ($n = 10$ per group). $p = 0.0400$ (*Chrebpβ*), $p = 0.0144$ (*Acly*), $p = 0.0405$ (*Acss2*), $p = 0.0297$ (*Fasn*) for comparison between Female Control and Female KO. **d** Proportion of assay for transposase-accessible chromatin (ATAC) peaks unique to each group of Control and KO mice. **e** Venn diagram showing the number of genes with reduced peak counts on ATAC-seq and reduced gene expression on RNA-seq in Female KO mice compared with Control mice. **f** Gene Ontology analysis performed on the 93 genes commonly downregulated in both ATAC-seq and RNA-seq. **g** Chromatin accessibility near the transcription start site (TSS) of *Chrebpβ* in the BAT of Control and KO mice. Data are expressed as the mean ± SEM. Data were analyzed by one-way ANOVA with Tukey's post hoc test. Significance is indicated (*$p < 0.05$). Source data are provided as a Source Data file.

genes through various histone modifications[20]. Therefore, to elucidate the female-specific role of PGC-1α in the transcriptional regulation of *Chrebpβ*, we conducted an assay for transposase-accessible chromatin using sequencing (ATAC-seq) on BAT from Control and KO mice of both sexes and evaluated chromatin accessibility across the genome.

Female Control mice exhibited more unique ATAC peaks than the other three groups (Fig. 3d), suggesting that PGC-1α played a role in maintaining specific chromatin states in female BAT. Integration of the ATAC-seq and RNA-seq data identified 93 genes with reduced chromatin accessibility and expression in Female KO mice compared with

Female Control mice (Fig. 3e). GO analysis of these genes highlighted pathways such as "monocarboxylic metabolic process," "adaptive thermogenesis," and "fatty acid metabolic process" (Fig. 3f). Furthermore, chromatin accessibility near the transcription start site (TSS) of *Chrebpβ* was higher in female BAT than in male BAT; however, this sex difference disappeared in PGC-1α KO mice (Fig. 3g), indicating that PGC-1α promoted *Chrebpβ* expression by maintaining open chromatin at the *Chrebpβ* locus. Although further studies are required to identify the molecular partners involved in PGC-1α−mediated histone modifications and the transcription factors that regulate *Chrebpβ* transcription in this setting, PGC-1α may play a critical role in regulating the chromatin accessibility and transcription of *Chrebpβ*, particularly in female BAT.

## PGC-1α deletion suppresses mitochondrial TCA cycle metabolism only in female BAT

To investigate the dynamics of energy substrate metabolism, we performed metabolomic analysis of central carbon metabolism in the BAT of Control and KO mice of both sexes. As shown in Fig. 4a, heat map analysis revealed marked differences between Male Control and Female Control mice, as well as between Female Control and Female KO mice. Among the 116 metabolites analyzed, there were 54 metabolites that were higher in Female Control mice than in Male Control mice but were lower in Female KO mice than in Female Control mice. Enrichment analysis of metabolic pathways showed that the citric acid cycle, ketones body metabolism, and butyrate metabolism pathways were significantly altered in Female KO mice compared with Female Control mice (Fig. 4b). Principal component analysis showed a large difference between Female Control and Female KO mice, whereas only a small difference between Male Control and Male KO mice was observed (Fig. 4c). The top 5 metabolites contributing positively to PC2, defining the difference between Female control and Female KO mice, include NADH and tricarboxylic acid (TCA) cycle intermediates such as fumaric acid, malic acid, and acetyl-CoA (Fig. 4d). Actually, TCA cycle intermediates were more abundant in Female Control mice than in Male Control mice and significantly decreased only in Female KO mice (Fig. 4e). In addition, in females, a positive correlation was observed between the levels of TCA cycle metabolites and the peak of $VO_2$ upon NE administration (Fig. 4f), suggesting that in female BAT, the activity of the TCA cycle may be a primary determinant of thermogenesis by acute adrenergic stimulation.

## PGC-1α deletion in female BAT modulates the profiles of membrane phospholipids, including cardiolipin and ether-linked phospholipids

DNL plays a key role in regulating lipid profiles in BAT[17,21]. Considering these findings, along with the mitochondrial morphological defects, functional impairments, and downregulation of DNL-related genes observed in the BAT of Female KO mice, we speculated that female PGC-1α may affect lipid profile in BAT. Therefore, we performed lipidomics in the BAT of Control and KO mice of both sexes. The analysis showed, first that the overall lipid profile of the BAT was significantly different between males and females, with females having less triglycerides and more abundant glycerophospholipids, which comprise the membranes of cells and organelles, compared with males (Fig. 5a). These results are consistent with those of a previous study of lipidomics in BAT of male and female wild-type mice[22]. Partial least squares (PLS) analysis also revealed that phospholipids with C > 17 fatty acids were more abundant in females than in males, and this pattern remained consistent even in PGC-1α KO mice (Figs. 5b, c and S4).

We next examined changes in cardiolipin (CL), a phospholipid specifically localized in the mitochondrial membrane and essential for regulating its morphology and respiratory capacity[23]. The percentage of total CL relative to BAT total lipids was higher in Female Control

mice than in Male Control mice; however, no reduction was observed in either sex of KO mice (Fig. 5d). Among the various CL species detected in BAT, the most abundant was the $CL(18:2)_4$ molecular species (Fig. 5e), which is also the most prevalent and functionally important species in the mammalian heart[24]. The percentage of this molecular species relative to total CL was significantly higher in Female Control mice than in Male Control mice, whereas it was significantly reduced only in Female KO mice (Fig. 5f). In addition, ether-linked phosphatidylethanolamine (PE) and phosphatidylcholine (PC), recently identified as critical lipids for maintaining mitochondrial morphology and function[25,26], were significantly more abundant in Female Control mice than in Male Control mice, whereas these phospholipids were substantially reduced only in Female KO mice (Fig. 5g, h). Furthermore, coenzyme Q9 (CoQ9) levels were significantly higher in Female Control mice than in Male Control mice and were reduced only in Female KO mice (Fig. 5i).

Taken together, the lipidomics revealed that sex differences exist in the phospholipid profiles of BAT and that female PGC-1α regulates the profiles of CL, ether-linked phospholipids, and CoQ9, all of which are crucial for mitochondrial respiratory function.

## ChREBPβ−mediated ether-linked PE synthesis regulates mitochondrial membrane structure and thermogenesis in female BAT

The aforementioned data suggest that in female BAT PGC-1α regulates thermogenesis by controlling the mitochondrial membrane structure and that ChREBPβ plays a pivotal role in this regulation. To further test this hypothesis, we performed ChREBP knockdown in the BAT of female mice by injecting adeno-associated viruses (AAV) containing short hairpin RNA (shRNA) targeting *Chrebp* (AAV-shChrebp) or scrambled shRNA (AAV-Scramble) into the interscapular BAT. One week after AAV injection, in the BAT of AAV-shChrebp mice, the expression of *Chrebpβ* and its downstream DNL-related genes was markedly reduced (Fig. 6a), whereas no significant changes were observed in the gene expression of *Chrebpα* (Fig. S5a). *Pgc1a* expression was significantly increased compared with that in AAV-Scramble mice (Fig. 6b). In addition, AAV-shChrebp mice displayed reduced BAT tissue weight and fat droplet size compared with AAV-Scramble mice (Fig. S5b, c). At this point, we found that $VO_2$ upon NE administration in the AAV-shChrebp mice was significantly lower than that in the AAV-Scramble mice (Fig. 6c). Electron microscopic examination of mitochondria showed that the total cristae length per mitochondrion was significantly shorter, and the size of mitochondria was also significantly reduced in the AAV-shChrebp mice compared with the AAV-Scramble mice (Figs. 6d, e and S5d). The results demonstrated that the phenotype observed only in female BAT due to PGC-1α deletion was recapitulated by ChREBP knockdown. Next, we analyzed the lipid profiles of AAV-Scramble and AAV-shChrebp female mice. The proportion of $CL(18:2)_4$ did not differ between the two groups (Fig. 6f). However, several ether-linked PEs were reduced in AAV-shChrebp mice (Fig. 6g), which was consistent with the observations in female PGC-1α KO mice (Fig. S5e). To investigate whether these ether-linked PEs were synthesized via DNL, we performed $D_2O$-labeling experiments in female Control and KO mice. PE(O-16:1_16:1), PE(O-16:1_18:1) and PE(O-18:1_16:0) were labeled by $D_2O$. Among the $D_2O$-labeled components of these ether-linked PEs, PE(O-16:1_18:1) was significantly reduced in KO mice (Fig. 6h). Overall, these results suggest that the de novo synthesis of ether-linked PE is mediated by the PGC-1α−ChREBPβ axis in female BAT.

In male BAT, AAV-shChrebp administration selectively reduced *Chrebpβ* expression without affecting *Chrebpα* expression (Fig. S6a, b). However, unlike in females, no significant changes in the mitochondrial membrane structure were observed (Fig. S6c−e). We analyzed the lipid profiles of male AAV-Scramble and AAV-shChrebp mice. The levels of $CL(18:2)_4$ and ether-linked PEs in the BAT of male AAV-shChrebp mice were not decreased compared with those in male AAV-

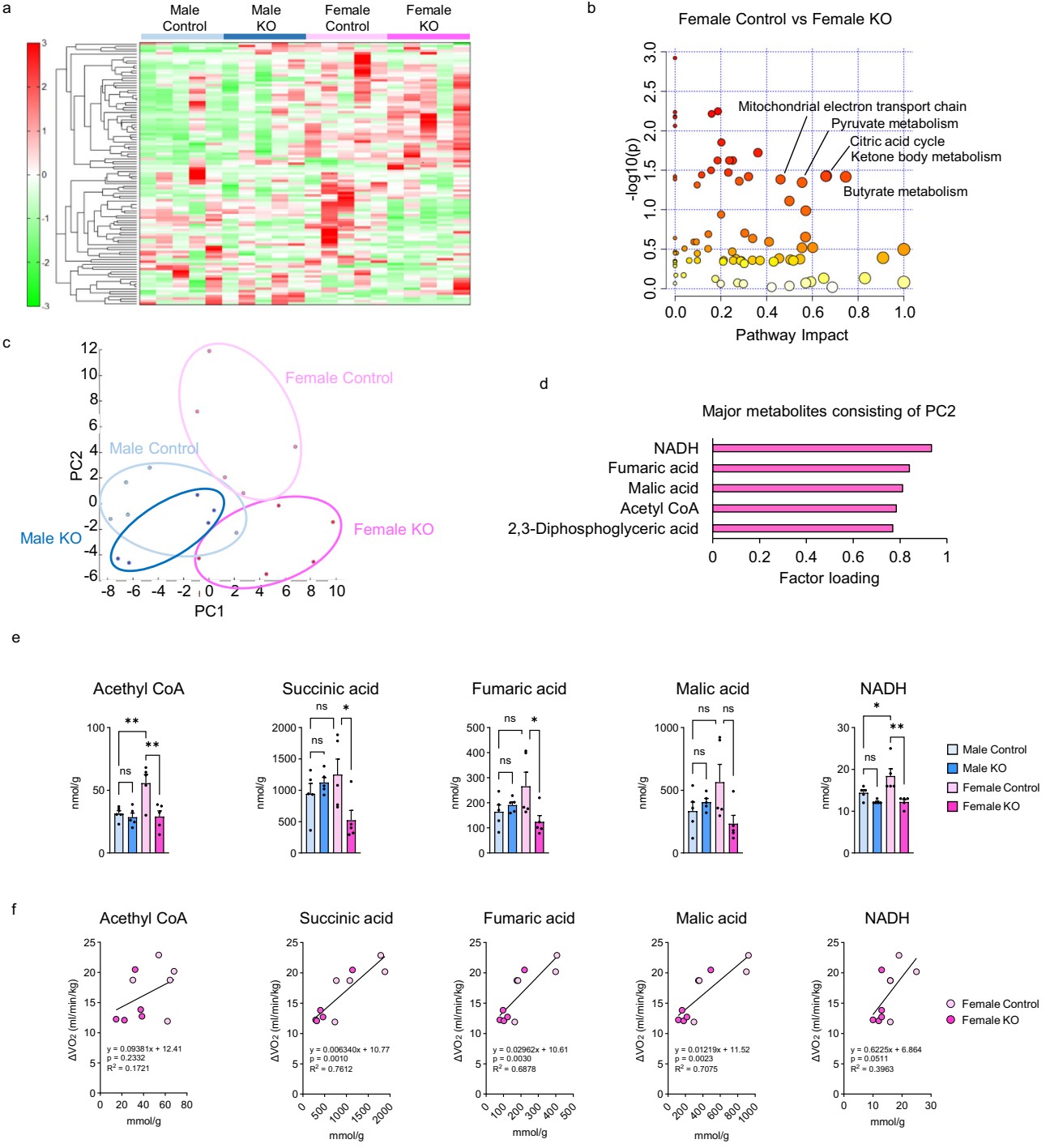

**Fig. 4 | PGC-1α regulates TCA cycle metabolism in female BAT. a** Heat map illustrating differential metabolites based on metabolomic analysis in the BAT of Control and KO mice of both sexes. $n = 5$ per group. **b** Enrichment analysis of metabolic pathways using MetaboAnalyst. The x-axis indicates the pathway impact values derived from the pathway topology analysis, and the y-axis indicates the −log10(p-values) obtained from the pathway enrichment analysis using the hypergeometric test (one-sided). P-values were adjusted for multiple comparisons using the false discovery rate (FDR) method. **c** Score plot of principal component analysis (PCA) of all metabolites. Each point represents an individual sample. **d** Top 5 metabolites contributing to

PC2 loading. **e** Concentration of TCA cycle metabolites in BAT of each group. $n = 5$ per group. $p = 0.0082$ and $0.0040$ for acetyl-CoA; $p = 0.6022$ and $0.0389$ for succinic acid; $p = 0.1974$ and $0.0445$ for fumaric acid; $p = 0.2551$ and $0.0575$ for malic acid; and $p = 0.0497$ and $0.0021$ for NADH, comparing Male Control vs. Female Control and Female Control vs. Female KO, respectively. **f** Correlation between the levels of TCA cycle metabolites and the peak of $VO_2$ upon NE administration. Data are expressed as the mean ± SEM. Data were analyzed by one-way analysis of variance with Tukey's post hoc test (**e**) or linear regression (**f**). Significance is indicated (*$p < 0.05$; **$p < 0.01$). Source data are provided as a Source Data file.

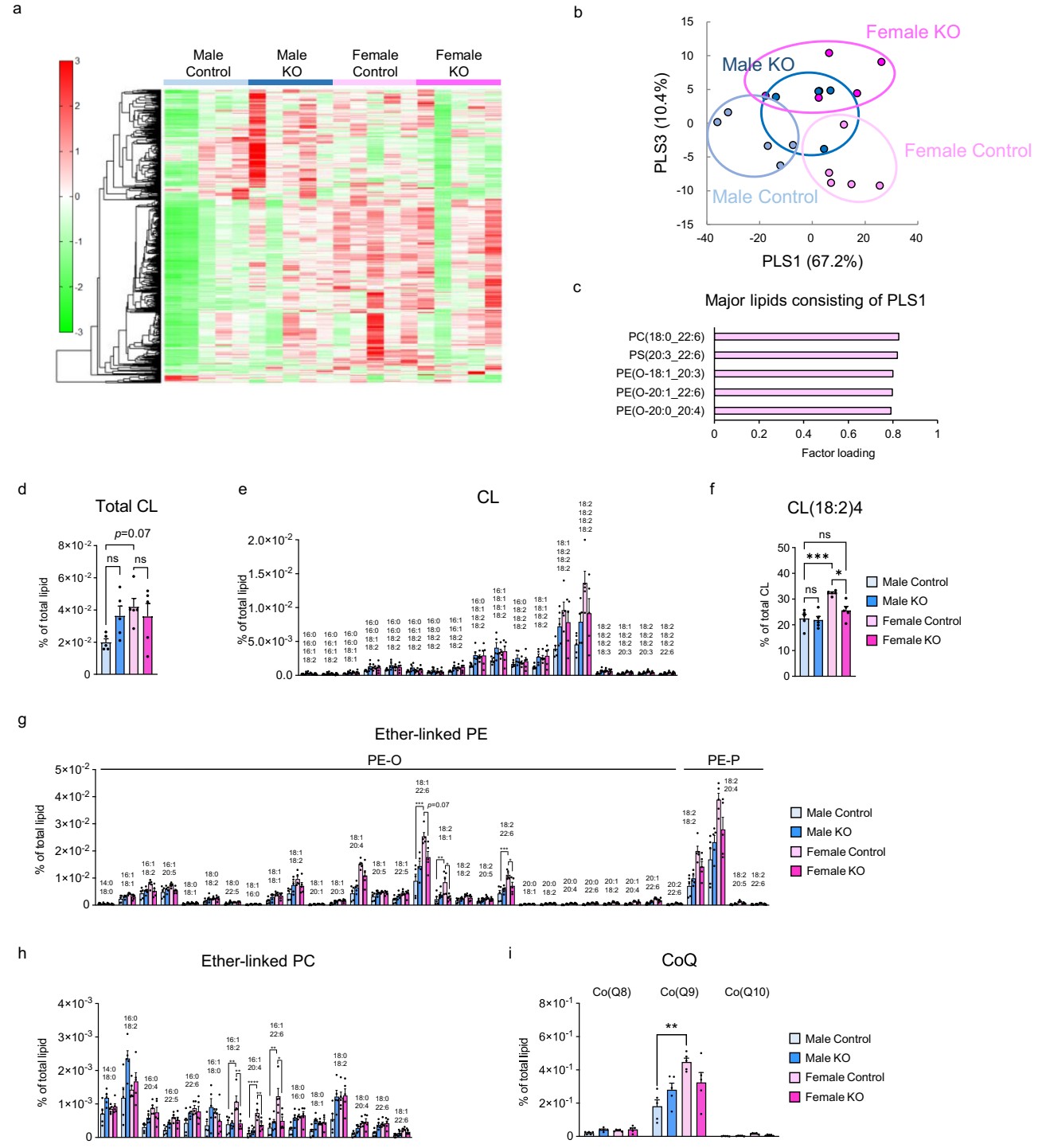

**Fig. 5 | PGC-1α changes the membrane phospholipid profiles of female BAT.**
**a** Heat map illustrating differential lipid species based on lipidomic analysis in the BAT of Control and KO mice of both sexes. $n = 5$ per group. **b** Score plot of partial least squares (PLS) discriminant analysis of all lipids. Each point represents an individual sample. **c** Top 5 lipids contributing to PLS1 loading. **d** Relative amounts of total cardiolipin (CL). $n = 5$ per group. **e** Various molecular species of cardiolipin. Numbers above the bar graph represent the molecular species composition. $n = 5$ per group. **f** Percentage of $CL(18:2)_4$ in total CL. $n = 5$ per group. $p = 0.0005$ and $0.0129$ for comparisons between Male Control vs. Female Control and Female Control vs. Female KO, respectively. Relative amounts of various molecular species of

ether-linked phosphatidylethanolamine (PE) (**g**) and phosphatidylcholine (PC) (**h**). $n = 5$ per group. **i** Relative amounts of coenzyme Q (CoQ). $n = 5$ per group. $p = 0.0002$ and $0.0747$ for PE-O(18:1/22:6), $p = 0.0059$ and $0.0123$ for PE-O(18:2/18:1), $p = 0.0008$ and $0.0477$ for PE-O(18:2/22:6), $p = 0.0051$ and $0.0064$ for PC-O(16:1/18:2), $p < 0.0001$ and $0.0085$ for PC-O(16:1/20:4), $p = 0.0022$ and $0.0149$ for PC-O(16:1/22:6), and $p = 0.0020$ and $0.2054$ for CoQ9, comparing Male Control vs. Female Control and Female Control vs. Female KO, respectively. Data are expressed as the mean ± SEM. Data were analyzed by one-way ANOVA with Tukey's post hoc test (**d**–**i**). Significance is indicated (*$p < 0.05$; **$p < 0.01$; ***$p < 0.001$; ****$p < 0.0001$). ns not significant. Source data are provided as a Source Data file.

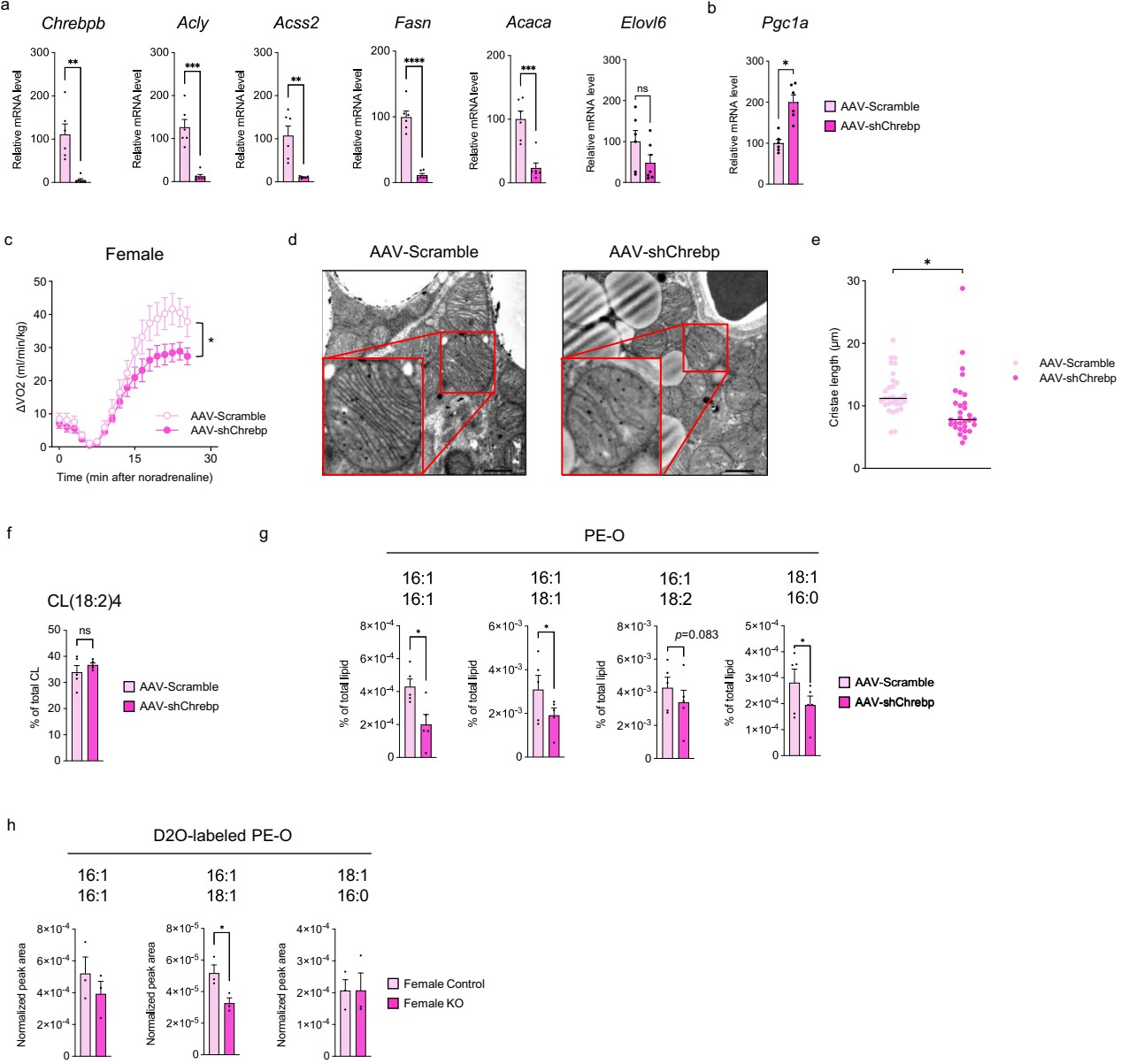

**Fig. 6 | BAT-specific ChREBPβ knockdown in female mice recapitulates the metabolic phenotype of female PGC-1α knockout mice. a** Gene expression of *Chrebpβ* and DNL-related genes in female BAT injected with adeno-associated viruses (AAV) -shScramble or AAV-shChrebp. $n = 6$ per group. $p = 0.0014$ for *Chrebpβ*, $p = 0.0001$ for *Acly*, $p = 0.0013$ for *Acss2*, $p < 0.0001$ for *Fasn*, $p = 0.0005$ for *Acaca*, and $p = 0.1527$ for *Elovl6*. **b** Gene expression of *Pgc1a*. $n = 6$ per group. $p = 0.0474$. **c** Oxygen consumption ($VO_2$) recordings in response to NE. $n = 7$ per group. $p = 0.0122$. **d** Representative electron micrographs of mitochondria from BAT of AAV-Scramble and AAV-shChrebp mice. Scale bar = 1 μm ($n = 3$ biologically independent experiments). **e** Total cristae length per mitochondrion. $n = 30$ per

group. $p = 0.0271$. **f** Percentage of CL(18:2)₄ in total CL. $n = 5$ per group. **g** Percentage of ether-linked PEs in total lipids. $n = 5$ per group. $p = 0.0236$, 0.0358, and 0.0395 for PE-O(16:1/16:1), PE-O(16:1/18:1), and PE-O(18:1/16:0), respectively. **h** $D_2O$-labeled components of ether-linked PEs. $n = 3$ per group. $p = 0.3849$, 0.0376, and 0.9923 for PE-O(16:1/16:1), PE-O(16:1/18:1), and PE-O(18:1/16:0), respectively. Data are expressed as the mean ± SEM. Data were analyzed using unpaired two-sided t-tests (**a**, **b**, **e**), paired one-sided t-tests (**f**–**h**), and two-way repeated measures ANOVA (**c**). Significance is indicated (*$p < 0.05$; **$p < 0.01$; ***$p < 0.001$; ****$p < 0.0001$). Source data are provided as a Source Data file.

Scramble mice (Fig. S6f, g), supporting the absence of alterations in the mitochondrial membrane structure.

**Estrogen signaling and PGC-1α interdependently regulate the expression of DNL-related genes in female BAT, thereby contributing to thermogenesis and mitochondrial membrane structure**

Because estrogen has previously been implicated in sex differences in BAT[27], we investigated the interaction between the female BAT-specific role of PGC-1α and estrogen signaling. We administered the estrogen receptor antagonist TMX or vehicle (Veh) to male and female wild-type

mice and measured $VO_2$ upon NE administration, 1 week after TMX treatment (Fig. S7a). In males, no difference was observed between the TMX-treated and Veh-treated mice, whereas in females, $VO_2$ was significantly lower in the TMX-treated mice than in the Veh-treated mice (Fig. 7a). Electron microscopic images of BAT showed that the length of mitochondrial cristae was significantly shortened by TMX treatment only in females (Fig. 7b, c). In addition, transcriptome analysis of BAT revealed that genes involved in energy substrate metabolism, especially in DNL-related genes, were downregulated by TMX treatment only in female mice (Fig. 7d–f). No significant changes were observed in *Chrebpα* expression in either sex (Fig. S7b), whereas total ChREBP

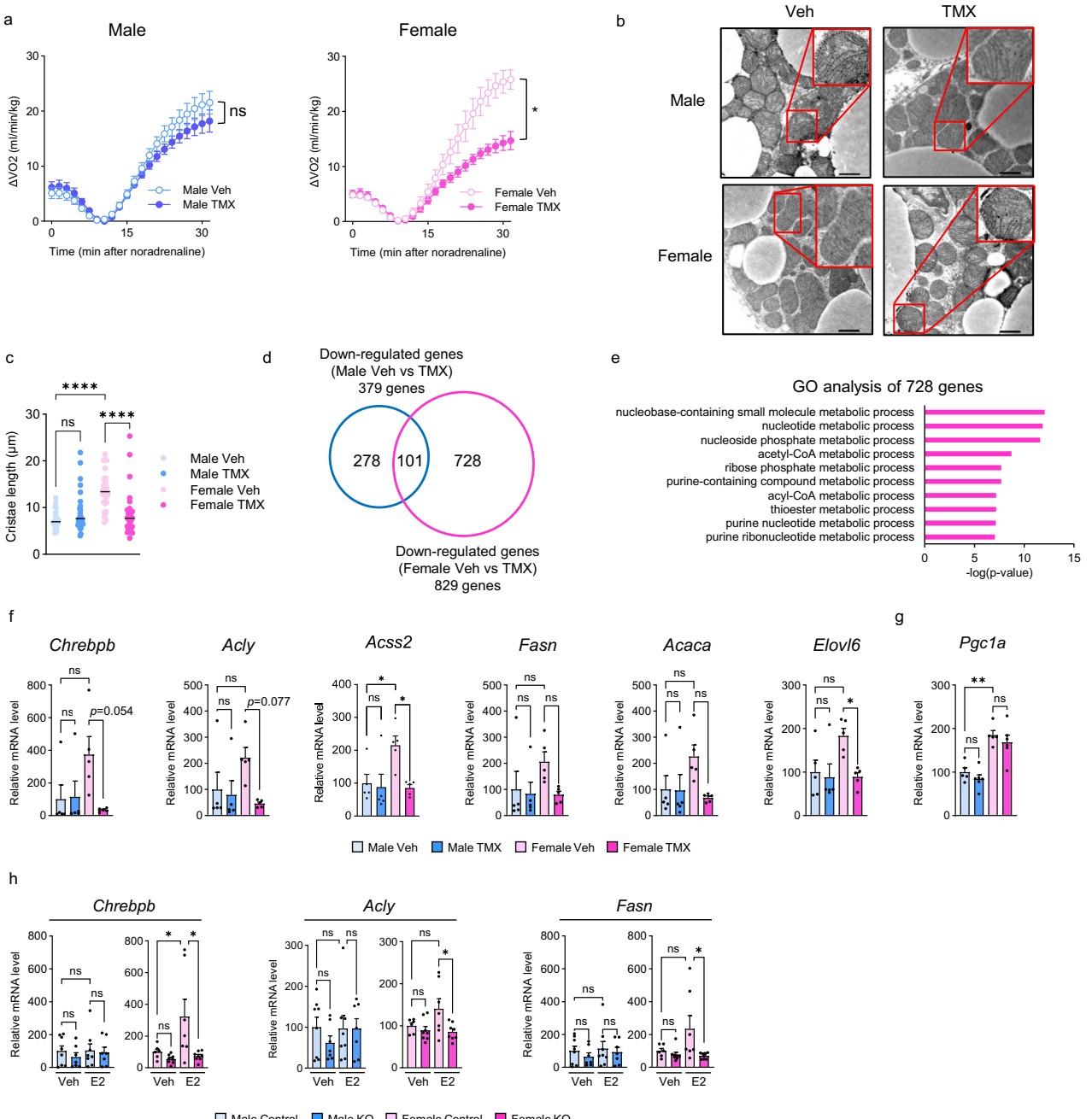

**Fig. 7 | Female BAT PGC-1α regulates systemic energy metabolism in coordination with estrogen signaling. a** Oxygen consumption (VO₂) recordings in response to NE in vehicle and tamoxifen (TMX) -treated mice. $n = 4$ per group. $p = 0.4129$ (Male Veh vs. Male TMX), $p = 0.0168$ (Female Veh vs. Female TMX). **b** Representative electron micrographs of mitochondria from BAT of control and TMX-treated mice. Scale bar = 1 μm. **c** Total cristae length per mitochondrion. $n = 30$ per group. $p < 0.0001$ for Male Veh vs. Female Veh, and $p < 0.0001$ for Female Veh vs. Female TMX. **d** Venn diagram showing the number of downregulated genes in male and female TMX-treated mice compared with controls. **e** Gene Ontology (GO) analysis of genes downregulated only in Female KO mice. The top 10 enriched terms are shown with enrichment p-values generated by Metascape using cumulative hypergeometric tests (one-sided). P-values were adjusted for multiple comparisons using the Benjamini-Hochberg procedure (FDR). **f** Gene expression of *Chrebpβ* and DNL-related

genes in BAT. $n = 5$ per group. $p = 0.0454$ and 0.0225 for Acss2 (Male Veh vs. Female Veh and Female Veh vs. Female TMX, respectively); $p = 0.0447$ for Elovl6 (Female Veh vs. Female TMX). **g** *Pgc1a* mRNA levels in BAT. $n = 5$ per group. $p = 0.0025$ (Male Veh vs. Female Veh). **h** Gene expression in BAT explants from the BAT of Control and KO mice of both sexes treated with vehicle ($n = 8$ Male Control, $n = 7$ Male KO, $n = 6$ Female Control, and $n = 8$ Female KO) or 17β-estradiol (E2) ($n = 8$ Male Control, $n = 7$ Male KO, $n = 7$ Female Control, and $n = 8$ Female KO). $p = 0.0465$ for Female Control Veh vs. E2, and $p = 0.0128$ for Female Control E2 vs. Female KO E2 for *Chrebpβ*; $p = 0.0339$ for Female Control E2 vs. Female KO E2 for *Acly*; $p = 0.0329$ for Female Control E2 vs. Female KO E2 for *Fasn*. Data are expressed as the mean ± SEM. Data were analyzed by two-way repeated measures ANOVA (**a**) and one-way ANOVA with Tukey's post hoc test (**c, f–h**). Significance is indicated (*$p < 0.05$; **$p < 0.01$; ***$p < 0.001$; ****$p < 0.0001$). ns not significant. Source data are provided as a Source Data file.

protein expression was significantly reduced only in female mice (Fig. S7c). Notably, in the PGC-1α KO experiments, an 8-week washout period was applied between TMX administration and analysis, as described earlier (Fig. S1e). During this period, the expression levels of

DNL-related genes in females, compared with those in males under the same conditions, returned to levels similar to those observed before TMX treatment (Fig. S7d). These results demonstrated that PGC-1α KO and TMX treatment share many metabolic phenotypes, histological

changes, and transcriptional alterations only in female BAT. However, TMX treatment did not downregulate *Pgc1a* expression in female BAT (Fig. 7g). A similar trend was observed in BAT gene expression, VO$_2$ after NE administration, and BAT mitochondrial morphology in ovariectomized mice compared with sham-operated mice (Fig. S7e–i). These findings suggest that ChREBPβ and its downstream DNL-related gene expression in female BAT are strongly regulated by estrogen signaling.

We additionally examined the effect of TMX on the sympathetic nervous system (SNS) output to BAT because estrogen has been shown to regulate BAT thermogenesis through SNS[28]. Immunolabeling of tyrosine hydroxylase (TH), a marker of SNS output, and its mRNA expression in BAT showed no significant differences between Veh-treated and TMX-treated mice in either sex (Fig. S8a, b). These results indicated that SNS output did not contribute to the observed alterations in the BAT of female TMX-treated mice.

### Estrogen signaling upregulates the expression of DNL-related genes in a PGC-1α-dependent manner only in female BAT

To investigate whether sex differences exist in the regulation of DNL-related gene expression in BAT by estrogen signaling, we performed an ex vivo experiment in which 17β-estradiol (E2) was applied to BAT explants from male and female wild-type mice. E2 treatment did not alter the expression of *Chrebpβ* and DNL-related genes in male BAT; however, it increased the expression of *Chrebpβ* and DNL-related genes in female BAT (Fig. S8c). In addition, the substantial increase in gene expression with E2 treatment observed in BAT from Female Control mice was attenuated in BAT from Female KO mice (Fig. 7h). To further investigate the mechanisms underlying sex differences in the PGC-1α dependent estrogen responsiveness, we examined the expression of *Esr1*, the gene encoding ERα, in the BAT of Control and KO mice of both sexes. We found that *Esr1* gene expression was significantly higher in female BAT than in male, but not affected by PGC-1α deletion in both sexes (Fig. S8d).

Collectively, these results suggest that the downstream molecule(s) of estrogen receptor (ERα) signaling pathway, which regulate *Chrebpβ* and DNL-related gene expressions, fundamentally differ between male and female BAT, and that PGC-1α plays a pivotal role in this difference.

## Discussion

Here, we showed that PGC-1α in BAT of female mice is highly expressed compared with that in male mice and plays a crucial role in regulating systemic energy expenditure by contributing to the maintenance of female BAT-specific cristae-rich mitochondria and high TCA cycle activity. Estrogen signaling interdependently regulates this mechanism with PGC-1α, and ChREBPβ plays a pivotal role as a central hub of both pathways.

In general, the role of PGC-1α in BAT has been recognized to promote thermogenesis through the upregulation of UCP1 expression[29,30]. However, a previous report of adipocyte-specific PGC-1α knockout mice revealed that its main phenotypes include insufficient induction of beige adipose tissue and impaired glucose tolerance, with limited changes in gene expression in BAT[11]. This report was based on male mouse model of congenital PGC-1α knockout, and there have been no investigations into the acquired knockout of PGC-1α in adult male and female adipose tissue. Thus, a distinct role of PGC-1α in female BAT may have been overlooked.

In our study, BAT *Pgc1a* was more highly expressed in females than in males under various conditions, particularly under cold exposure and upon adrenergic stimulation, and its gene expression positively correlated with that of *Adcy3*. In addition, adipocyte-specific PGC-1α deletion resulted in a lower body temperature during acute cold exposure and suppression of VO$_2$ upon NE administration only in female mice. These data suggest that PGC-1α in BAT of female mice is

regulated by strong adrenergic stimulation and contributes to an efficient response to rapidly increasing energy demand.

Recently, it was reported that PGC-1α has female-specific functions in some organs. For example, PGC-1α regulates oxidative stress responses in coordination with estrogen signaling in the liver of female mice[31] and intracellular and extracellular calcium dynamics in the female myocardium[32]. Considering the results of these reports and our study, PGC-1α may be a molecule that mediates sex differences across multiple organs, especially the characteristics of females.

In this study, we found a distinctive mitochondrial morphology in female BAT, which was characterized by a significantly longer total cristae length and larger size than that in male mice. Moreover, not only in terms of morphology but also in functional aspects, mitochondria in female BAT were found to exhibit high NADH-producing capacity and TCA cycle activity. In previous reports, the significance of TCA cycle metabolism in BAT thermogenesis has been demonstrated[33,34]. We have also shown that these morphological and functional features of BAT in female mice are generated by PGC-1α.

CL is a phospholipid that is specific to mitochondrial membranes and is essential for the maintenance of mitochondrial structure, respiratory capacity, and energy substrate metabolism, including the TCA cycle[23,35–37]. CL has also been reported to be a strong regulator of BAT thermogenesis[38]. In this study, we found that the percentage of $CL(18:2)_4$, which is especially important for mitochondrial respiration[23,24], was higher in females than in males. Notably, PGC-1α deletion significantly reduced the percentage of $CL(18:2)_4$ in females, whereas ChREBPβ knockdown had no such effect. These findings suggest that in females, PGC-1α specifically regulates CL lipid profiles via a mechanism independent of ChREBPβ. It was reported that cardiomyocyte PGC-1α/β regulated total CL content and mitochondrial morphology by regulating CDP-diglyceride synthetase 1(CDS1) gene expression, an enzyme in the phospholipid synthesis pathway[39]. However, in our study, *Cds1* gene expression in BAT showed little change in both males and females due to PGC-1α deletion (Fig. S3d), suggesting the existence of a tissue-specific molecular mechanism(s) in PGC-1α–mediated CL production.

A recent study reported that ether-linked PEs and PCs promote the formation of the electron transport chain supercomplex and maintain mitochondrial reactive oxygen species homeostasis in vitro[25]. Moreover, another study reported that alterations in ether lipid profile, such as those of ether-linked PEs and PCs, lead to mitochondrial morphological abnormalities in cell lines[26]. Notably, ether-linked PEs were reduced in our lipidomics following both PGC-1α KO and ChREBPβ knockdown. Overall, PGC-1α may play a pivotal role in mitochondrial morphology and function in female BAT through ChREBPβ-dependent ether-linked PE synthesis and ChREBPβ-independent $CL(18:2)_4$ synthesis.

Although female mitochondria reportedly exhibit a higher energy-producing capacity than male mitochondria in several mitochondria-rich organs, such as the liver, brain, and skeletal muscle[15], studies investigating the underlying molecular mechanism are limited. While little is known about sex differences in TCA cycle metabolism, a previous study analyzing human brain samples reported that the activities of enzymes involved in the TCA cycle, such as succinate dehydrogenase and citrate synthase, were higher in females than in males[40]. In addition, a previous study reported that female rats had a higher total CL amount in the liver than males[41,42] but did not analyze the fatty acid profile of CL. Considering these findings, it is probable that sex differences in the fatty acid profiles of CL and their regulatory mechanisms exist in organs/tissues beyond BAT.

We observed that estrogen receptor antagonist administration suppressed VO$_2$, impaired mitochondrial structure, and reduced expression of *Chrebpβ* and downstream DNL-related genes only in BAT of female mice without reducing *Pgc1a* expression. These results suggest that estrogen signaling and PGC-1α interdependently regulate

mitochondrial structure and thermogenesis in female BAT, and we further demonstrated that the transcriptional regulation of *Chrebpβ* serves as a central hub in coordinating these two regulatory pathways. There are limited reports on the regulation of lipogenesis by PGC-1-α[43], while the association between lipogenesis and estrogen has been reported in various cells and organs. In estrogen-sensitive breast and gynecological cancers, estrogen signaling promoted lipogenesis, which in turn contributed to cell proliferation[44,45]. Given the high estrogen sensitivity of female BAT, these estrogen-sensitive cancer cells may have mechanisms similar to those of female BAT for maintaining its high metabolic activity.

A previous report has shown that estrogen signaling in the central nervous system enhances BAT thermogenesis in female rats by increasing sympathetic nervous system activation[28]. On the other hand, our BAT explant analyses indicated that the regulatory system of estrogen, PGC-1α, and DNL-related genes in female BAT is tissue-autonomous. Taken together, the previous findings and our results suggest that estrogen signaling may enhance BAT thermogenesis in direct and indirect manners.

Based on previous reports that serum estrogen levels are extremely lower in male mice compared to female[46], the strong involvement of PGC-1α in the estrogen–lipogenesis regulatory mechanism in female BAT may provide one explanation for why the deletion of PGC-1α in male BAT does not result in the same phenotype observed in female BAT. On the other hand, even considering the sex differences in *Esr1* expression levels, the absence of an upregulation in DNL-related gene expression in male BAT explants upon direct E2 supplementation suggests that the estrogen–lipogenesis regulatory system itself does not exist in male BAT. The previous report of no enhancement in BAT thermogenesis in male rats following peripheral estrogen administration[47] aligns with our results of the ex vivo experiment. In addition, it has been reported that male-to-female transsexuals undergoing estrogen replacement therapy exhibit a tendency toward obesity[48]. Estrogens have been reported to potentially increase body fat through mechanisms such as the proliferation of preadipocyte[49]. Therefore, estrogen administration to men may increase WAT abundance but may not enhance BAT energy expenditure, which is one possible reason why men cannot benefit from the estrogen that women receive.

In summary, our findings demonstrate that PGC-1α in female BAT plays a crucial role in controlling systemic energy expenditure by regulating the distinct morphology and function of mitochondria in coordination with estrogen signaling. While PGC-1α has been generally recognized as a master regulator of mitochondria, we have elucidated a novel mitochondrial regulatory mechanism of PGC-1α specific to female BAT. Given the inverse correlation between BAT and metabolic or cardiovascular disease, it is likely that this mechanism could contribute to the metabolic advantage in females. PGC-1α and estrogen signaling are interdependent in this system, meaning that the absence of either one severely impairs thermogenesis function. This "over-dependence" of female BAT on estrogen may, in turn, explain why postmenopausal women rapidly lose their metabolic advantage, and thus may be a new therapeutic target for metabolic diseases in postmenopausal women.

## Methods
### Animals
All animal experiments were approved by the Institute of Science Tokyo Committee on Animal Research (approval numbers: A2023-020A, G2023-073A), and were conducted in accordance with the Fundamental Guidelines for Proper Conduct of Animal Experiment and Related Activities in Academic Research Institutions under the jurisdiction of the Ministry of Education, Culture, Sports, Science and Technology of Japan. C57BL/6J wild-type mice were purchased from CLEA Japan, Inc., and *Adipoq*-CreERT2 (stock number: 025124) and

*Pgc1a* flox/flox (stock number: 009666) mice were purchased from The Jackson Laboratory. To generate tamoxifen-inducible adipocyte-specific PGC-1α knockout (KO) mice, *Pgc1a* flox/flox mice were crossed with *Adipoq*-CreERT2 mice. The animals were housed alone at 25 °C with a 12-h light/dark cycle and allowed free access to water and a standard diet (CE-2; 343 kcal/100 g, CLEA Japan, Inc.). Eight-week-old male and female KO or *Pgc1a* flox/flox (Control) mice were injected intraperitoneally with 100 mg/kg tamoxifen (Sigma-Aldrich) for 5 consecutive days and analyzed at least 8 weeks after the last administration of tamoxifen. For the experiment of tamoxifen administration to wild-type mice according to the same protocol, and analyzed after 2 weeks.

For cold exposure experiments, mice were maintained at thermoneutrality (30 °C) in a temperature-controlled chamber (HC-100, Shin Factory, Japan) for 1 week before cold challenge and allowed free access to water and a standard diet. Food was removed 1 h before the start of cold exposure, and the animals were placed at 4 °C. During cold exposure, the rectal temperatures (TD-300, NATSUME, Japan) and interscapular surface temperatures (E53, FLIR Systems Inc., USA) of mice were measured every 30–60 min. The method for measuring interscapular surface temperatures is described in detail in a previous report[50]. At the end of each experiment, mice were euthanized by cervical dislocation in accordance with institutional guidelines.

### NE-induced oxygen consumption
Oxygen consumption was measured using a metabolic chamber (Shin Factory, JAPAN) coupled to a mass spectrometer (ARCO-2000; Arco system, Tokyo, JAPAN). Mice were anesthetized, and measurements were performed for 30 min at 33 °C to obtain basal values. Each mouse was then briefly removed from the chamber, treated with norepinephrine (1 mg norepinephrine/kgBW), and returned to the chamber, and oxygen consumption was measured for another 30–40 min.

### Glucose tolerance test
The glucose tolerance test was performed via an intraperitoneal injection of glucose at 2.0 g/kg body weight, and blood glucose levels were measured before and 15, 30, 60, 90, and 120 min after the injection. Blood glucose was measured using a glucometer (Stat Strip Xpress; Nova Biomedical, USA).

### Western blotting
Protein lysates were extracted from Brown adipose tissue using RIPA buffer (Nacalai) supplemented with a protease inhibitor cocktail (cOmplete™, Sigma-Aldrich). Immunoblotting was performed with Anti-PGC-1α Mouse mAb (Sigma-Aldrich, ST1202, 1:1000), Rabbit polyclonal ChREBP antibody (Novus Biologicals, NB400-135, 1:1000), and total OXPHOS rodent WB antibody cocktail (Abcam, ab110413, 1:1000). α-tublin (Cell Signaling, #2144, 1:1000) was used as the loading control. Immunoblots were detected and analyzed using ECL Prime Western Blotting Detection Reagent and ImageQuant LAS 4000 mini (GE Healthcare).

### HE staining
BAT was fixed with 4% paraformaldehyde and embedded in paraffin. Sections were stained with hematoxylin and eosin (HE).

### Immunolabeling of tyrosine hydroxylase
BAT was fixed with 4% paraformaldehyde and embedded in paraffin. After deparaffinization, antigen retrieval was performed using the Target Retrieval Solution (Agilent Dako, S1699). Endogenous peroxidase activity was blocked with 0.3% $H_2O_2$ in methanol. Blocking was performed using 5% normal goat serum/0.5% BSA in PBS (Normal Goat Serum Blocking Solution, Vector Laboratories, S-1000). The primary antibody (Tyrosine Hydroxylase (E2L6M) Rabbit mAb, Cell Signaling, #58844), diluted 1:100 in PBS containing 0.5% BSA, was applied and

incubated overnight at 4 °C. The secondary antibody, N-Histofine Simple Stain Mouse MAX-PO(R) (Nichirei Biosciences, 414341), was applied and incubated at room temperature for 1 h. Color development was performed using DAB (Dako, K3468).

## Electron microscopy

The BAT were fixed in 4% PFA and 2.5% GA in 0.1 M phosphate buffer (PB) for 2 h, washed with 0.1 M PB, post-fixed in 1% $OsO_4$ buffered with 0.1 M PB for 2 h, dehydrated in a graded series of ethanol, and embedded in Epon 812. Ultrathin sections (70 nm) were collected on copper grids, double-stained with uranyl acetate and lead citrate, and then examined by transmission electron microscopy (JEM-1400Flash, JEOL, Japan). Quantification of mitochondrial size and content was performed using ImageJ software.

## RNA isolation and quantitative RT-PCR

Total RNA was isolated using the RNeasy Plus Universal Mini Kit (Qiagen). cDNA was synthesized using Random Primer (Thermo Fisher Scientific Inc.) and ReverTra Ace (Toyobo Co., Ltd.). Quantitative PCR was performed using the QuantStudio 6 Flex Real-Time PCR System with Fast SYBR Green Master Mix Reagent. The primer sequences are presented in Supplementary Table 1.

## RNA sequencing

RNA-seq experiments were performed by Novogene (Beijing, China) using RNA extracted from BAT. Sequencing libraries were built using the NEBNext UltraTM RNA Library Prep Kit (Illumina, USA). The library preparations were sequenced on an Illumina Novaseq 6000 platform, and 150-bp paired-end reads were generated. Differentially expressed genes were determined by fold change (>1.5), and gene ontology analysis was conducted using Metascape 3.5.

## ATAC-seq

Flash-frozen tissues were sent to Active Motif to perform the ATAC-seq assay. The tissue was manually disassociated, isolated nuclei were quantified using a hemocytometer, and 100,000 nuclei were tagmented as previously described (Buenrostro et al.[51]), with some modifications based on (Corces et al.[52]) using the enzyme and buffer provided in the ATAC-Seq Kit (Active Motif). Tagmented DNA was then purified using the MinElute PCR purification kit (Qiagen), amplified with 10 cycles of PCR, and purified using Agencourt AMPure SPRI beads (Beckman Coulter). The resulting material was sequenced with PE42 sequencing on the NovaSeq 6000 sequencer (Illumina).

Reads were aligned using the Burrows-Wheeler Aligner (BWA) algorithm (mem mode; default settings). Duplicate reads were removed, and only reads mapping as matched pairs and only uniquely mapped reads (mapping quality ≥ 1) were used for further analysis. Alignments were extended in silico at their 3′-ends to a length of 200 bp and assigned to 32-nt bins along the genome. The resulting histograms (genomic "signal maps") were stored in bigWig files. Peaks were identified using the MACS 3.0.0 algorithm at a cutoff of p-value 1e-7, without the control file, and with the −nomodel option. Peaks on the ENCODE blacklist of known false ChIP-Seq peaks were removed. Signal maps and peak locations were used as input data for the Active Motifs proprietary analysis program, which creates Excel tables containing detailed information on sample comparison, peak metrics, peak locations, and gene annotations. Tracks were visualized using the UCSC Genome Browser (https://genome.ucsc.edu/).

## Metabolomics

A total of 20 BAT samples for metabolomics ($n = 5$ per group; Male Control, Male KO, Female Control, and Female KO) were obtained from mice 30 min after NE administration at 33 °C to evaluate the metabolic dynamics of BAT exhibiting maximal oxygen consumption. Approximately 25–30 mg of frozen BAT tissue was placed in a homogenization tube along with zirconia beads (5mmφ and 3mmφ). Next, 1,500 μL of 50% acetonitrile/Milli-Q water containing internal standards (H3304-1002, Human Metabolome Technologies, Inc. (HMT), Tsuruoka, Yamagata, Japan) was added to the tube, after which the tissue was completely homogenized at 1500 rpm, 4 °C for 60 s using a bead shaker (Shake Master NEO, Bio Medical Science, Tokyo, Japan). The homogenate was then centrifuged at $2300 \times g$, 4 °C for 5 min. Subsequently, 800 μL of upper aqueous layer was centrifugally filtered through a Millipore 5-kDa cutoff filter (UltrafreeMC-PLHCC, HMT) at $9100 \times g$, 4 °C for 180 min to remove macromolecules. The filtrate was evaporated to dryness under vacuum and reconstituted in 50 μL of Milli-Q water for metabolome analysis at HMT.

Metabolome analysis was conducted according to HMT's C-SCOPE package, using capillary electrophoresis time-of-flight mass spectrometry (CE-TOFMS) for cation analysis and CE-tandem mass spectrometry (CE-MS/MS) for anion analysis based on the methods described previously[53,54]. Briefly, CE-TOFMS and CE-MS/MS analyses were performed using an Agilent CE capillary electrophoresis system equipped with an Agilent 6210 time-of-flight mass spectrometer (Agilent Technologies, Inc., Santa Clara, CA, USA) and Agilent 6460 Triple Quadrupole LC/MS (Agilent Technologies), respectively. The systems were controlled using Agilent G2201AA ChemStation software version B.03.01 for CE (Agilent Technologies) and connected by a fused silica capillary (50 μm i. d. × 80 cm total length) with commercial electrophoresis buffer (H3301-1001 and I3302-1023 for cation and anion analyses, respectively, HMT) as the electrolyte. The time-of-flight mass spectrometer was scanned from m/z 50 to 1000[53], and the triple quadrupole mass spectrometer was used to detect compounds in dynamic MRM mode. Peaks were extracted using MasterHands, automatic integration software (Keio University, Tsuruoka, Yamagata, Japan)[55] and MassHunter Quantitative Analysis B.04.00 (Agilent Technologies) to obtain peak information, including m/z, peak area, and migration time (MT). Signal peaks were annotated according to the HMT metabolite database based on their m/z values and MTs. The peak area of each metabolite was normalized to internal standards, and the metabolite concentration was evaluated by standard curves with three-point calibrations using each standard compound. Hierarchical cluster analysis and principal component analysis (PCA)[56] were performed using HMT's proprietary MATLAB and R programs, respectively. Detected metabolites were plotted on metabolic pathway maps using the VANTED software[57].

## Partial least squares (PLS) discriminant analysis

Metabolomics data were normalized and analyzed by partial least squares (PLS)[58] using R programs[59] developed by Human Metabolome Technologies, Inc.

## Lipidomics

A total of 20 BAT samples for lipidomics ($n = 5$ per group; Male Control, Male KO, Female Control, and Female KO) were obtained from mice 30 min after NE administration at 33 °C to evaluate the lipid profile of BAT exhibiting maximal oxygen consumption. Lipidome analysis was conducted according to lipidome lab Non-targeted lipidome Scan package (lipidome lab, Akita, Japan), using liquid chromatograph orbitrap mass spectrometry (LC-OrbitrapMS) based on the methods described previously[60,61]. Briefly, total lipids were extracted from 1 mg brown adipose tissue samples with the modified Bligh-Dyer method. An aliquot of the lower/organic phase was evaporated to dryness under N2, and the residue was dissolved in methanol for LC-MS/MS measurements.

Liquid chromatography (LC)-electrospray ionization-MS/MS analysis was performed by using Q-Exactive Plus mass spectrometer with an UltiMate 3000 LC system (Thermo Fisher Scientific). Samples were separated on L-column3 C18 metal-free column (2.0 μm, 2.0 × 100 mm i.d.) at 40 °C using a gradient solvent system: mobile phase A

(isopropanol/methanol/water (5/1/4 v/v/v) supplemented with 5 mM ammonium formate and 0.05% ammonium hydroxide (28% in water))/ mobile phase B (isopropanol supplemented with 5 mM ammonium formate and 0.05% ammonium hydroxide (28% in water)) ratios of 60%/ 40% (0 min), 40%/60% (0–1 min), 20%/80% (1–9 min), 5%/95% (9–11 min), 5%/95% (11–22 min), 95%/5% (22–22.1 min), 95%/5% (22.1–25 min), 60%/40% (25–25.1 min) and 60%/40% (25.1–30 min). The injection volume was 10 μl, and the flow rate was 0.1 mL/min. A heated electrospray ionization (HESI-II) source conditions were as follows: ionization mode, positive or negative; sheath gas, 60 arbitrary units; auxiliary gas, 10 arbitrary units; sweep gas, 0 arbitrary units; spray voltage, 3.2 kV in positive and −3.0 kV in negative mode; heater temperature, 325 °C; ion transfer capillary temperature, 300 °C in positive and −320 °C in negative mode; and S-lens RF level, 50. The orbitrap mass analyzer was operated at a resolving power of 70,000 in full-scan mode (scan range 200–1800 $m/z$ in positive and 190–1800 $m/z$ in negative mode; automatic gain control (AGC) target 1e6 in positive and 3e6 in negative mode) and resolving power of 17,500 in positive and 35,000 in negative mode in the top 20 data-dependent MS2 mode (stepped normalized collision energy 20, 30 and 40; isolation window 4.0 m/z; AGC target 1e5) with dynamic exclusion setting of 10.0 s. Postprocessing of the raw data files for diacylglycerol and ceramide were done using the lipid molecular identification software, Lipid Search 5.1 (Mitsui Knowledge Industries Co., Ltd, JAPAN), which identifies individual intact lipid molecules on the basis of their molecular weight and fragmentation patterns from headgroup and fatty acid composition. In this method, biological matrix effects cannot be normalized in all detected peaks, because it is not possible to prepare appropriate internal standards corresponding to all the detected peaks. The relative values were calculated using the ratio of the chromatographic peak area of each analyte to that of the total analyte. The annotation method used in this study corresponds to equivalent to "Fatty Acyl/Alkyl Level or Hydroxyl Group Level" defined by the lipidomics Standard Initiative[62].

### D₂O-labeling and lipid analysis

Three days prior to tissue collection, mice were intraperitoneally injected with 0.035 mL/g body weight of 0.9% NaCl $D_2O$ (Sigma, 151882). The drinking water was replaced with 8% $D_2O$-enriched water for the duration of the labeling period.

Total lipids were extracted from the 6 BAT samples ($n = 3$ per group; Female Control and Female KO) using the modified Bligh-Dyer method. The organic phase was evaporated to dryness under N2 and treated with hexane/methanol (2:1, v/v) to remove neutral lipids. The sample was then dried again, and the residue was dissolved in methanol/isopropanol (1:1, v/v) containing an internal standard (PE (12:0_12:0)) for LC-MS/MS analysis.

Liquid chromatography (LC)-electrospray ionization-MS/MS analysis was performed on a Shimadzu Nexera X3 ultra-high-performance LC system (Shimadzu) coupled to a QTRAP 7500 hybrid triple quadrupole linear ion trap mass spectrometer (AB SCIEX). Samples were separated on an Acquity UPLC HSS T3 column (100 × 2.1 mm, 1.8 μm; Waters) at 40 °C with a mobile phase consisting of solvent A (water/ methanol (1:1, v/v) containing 10 mM ammonium acetate and 0.2% acetic acid) and solvent B (isopropanol/acetone (1:1, v/v)) under a gradient program (0–3 min: 30% B → 50% B; 3–24 min: 50% B → 90% B; 24–28 min: 30% B). The injection volume was 10 μl, and the flow rate was 0.3 mL/min. The instrument parameters were as follows: curtain gas, 32 psi; collision gas, 8 arbitrary units; ionspray voltage, −4500 V; temperature, 300 °C; ion source gas 1, 40 psi; ion source gas 2, 60 psi. The amounts of deuterium-labeled PE were quantified by multiple reaction monitoring (MRM). The MRM transitions were as follows: m/z 672.5 → 256.3 for PE (O-16:1_d3-16:1), m/z 705.5 → 286.0 for PE (O-16:1_d5-18:1), m/z 701.5 → 258.0 for PE (O-18:1_d3-16:0), and m/z 578.5 → 199.0 for PE (12:0_12:0). The data were acquired and processed using SciexOS software (version 3.3.1.43; AB SCIEX). The peak area of each PE species was normalized to internal standard.

### BAT-specific ChREBP knockdown by AAV injection

AAV vectors expressing shRNA that targets Chrebp (Mlxipl, target sequence: GGACTGCTTCTTGTCCGATAT) and scrambled shRNA were obtained from VectorBuilder VB230122-1164ver and VB010000-0023jze, respectively. Wild-type female mice were anesthetized and the interscapular skin was incised to expose the BAT. $2.4 \times 10^{10}$GC vectors were injected into the BAT using 10 μL syringe (HAMILTON, 80330), the skin was sutured, and the anesthesia was antagonized. These mice were used for experiments of measurement of $VO_2$, gene expression analyses, and histological analyses after 1 week of injection.

### Ovariectomy

12-week-old wild-type mice were bilaterally ovariectomized (OVX) or sham operated under a combination of medetomidine, midazolam, and butorphanol anesthesia (subcutaneously administered). They were sacrificed and analyzed 1 week after ovariectomy.

### BAT ex vivo assay

BAT was collected, surface washed with PBS, and kept in PBS at 37 °C. BAT explants were cut into 1–2 mm pieces with scissors in PBS at 37 °C and transferred to serum and phenol red-free high glucose DMEM supplemented with 100 nM 17β-Estradiol (Sigma) or vehicle (ethanol). After incubation in a $CO_2$ incubator for 24 h, BAT was collected and used for RNA extraction.

## Data availability

RNA-seq and ATAC-seq data are available in NCBI's Gene Expression Omnibus under accession numbers GSE288721 and GSE288779. Metabolome analysis data are available in the Metabolomics Workbench under the accession number ST003923. Source data are provided with this paper. All data supporting the findings described in this manuscript are available in the article and in the Supplementary Information and from the corresponding author upon request. Source data are provided with this paper.

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

## Acknowledgements

We thank Ms. K. Katakura and T. Haba for technical assistance. This work was supported by a Grant-in-Aid for Scientific Research to T.Y. (22H03126) and K.T. (21K16350) from the Japan Society for the Promotion of Science (JSPS). This research was also supported by Moonshot R&D [Grant Number JPMJPS2023] to T.Y. from the Japan Science and Technology Agency (JST).

## Author contributions

A.T., J.A., and K.T. conceived the project, designed and performed the experiments, and evaluated the data. T.Y. conceived the project and designed the experiments. K.I., Y.N., M.H., K.H., R.O., R.K., M.M., K.S., and C.K. contributed to the discussion. R.O. contributed to the immunolabeling of tyrosine hydroxylase. N.K., K.K., and Junken A. contributed to the analysis of $D_2O$-labeled lipids. A.T., K.T., and T.Y. contributed to the discussion and wrote, reviewed, and edited the manuscript.

## Competing interests

The authors declare no competing interests.
