## [Transparent Peer Review file · Nature Communications]

Sex difference in BAT thermogenesis depends on PGC-1 α -mediated phospholipid synthesis in mice

Corresponding Author: Dr Kazutaka Tsujimoto

Version 0:

Reviewer comments:

Reviewer #1

(Remarks to the Author)

In this manuscript, the authors determined the role of transcriptional coactivator PGC-1 α in sex-specific regulation of brown adipose tissue (BAT) thermogenic activity by investigating adipocyte-specific PGC-1 α knockout (AKO) male and female mice. They found that PGC-1 α AKO females exhibit less cold tolerance along with reduced BAT temperature and lower response to norepinephrine compared to males. This reduction in BAT thermogenesis was associated with altered mitochondrial membrane structure and lipid profiles, and decreased TCA cycle metabolites. In addition, expression of genes involved in ChREBP β -mediated de novo lipogenesis (DNL) pathway was reduced in PGC-1 α AKO females but not in PGC-1 α AKO males. Based on these findings, the authors conclude that PGC-1 α plays a unique, sex-specific role in BAT of female mice through activation of ChREBP β -mediated DNL, which contributes to the maintenance of mitochondrial membrane structure. However, it is not clear how PGC-1 α promotes ChREBP β -mediated DNL. In addition, although the authors concluded that estrogen signaling also regulates ChREBP β /DNL-related gene expression in female BAT, the relative contribution of direct estrogen signaling vs increased sympathetic nervous system (SNS) output to enhancing this pathway in BAT remains unclear. It is likely that PGC-1 α is one of many downstream factors activated in BAT of female mice due to BAT activation via enhanced SNS output.

1. It is not clear how PGC-1 α promotes ChREBP β -mediated DNL. Does PGC-1 α interact with and coactivate ChREBP β ? PGC-1 β has been shown to coactivate ChREBP β and promote DNL gene expression. Does PGC-1 α deletion alter PGC-1 β expression?
2. It is not clear whether AAV-shChrebp targets specifically Chrebbp without altering Chrebpa expression. What is the effect of ChREBP β knockdown on mitochondrial membrane structure and lipid profiles in male BAT?
3. Estrogen has been shown to enhance BAT function by increasing sympathetic nervous system (SNS) output to BAT (PMC4082097). In addition, mild cold (room temp)-induced AKT2 signaling has been shown to stimulate ChREBP-mediated DNL (PMC5762420). These findings indicate that increased SNS output to BAT in female mice can stimulate PGC-1 α expression and ChREBP-mediated DNL. Estrogen receptor antagonist TMX is likely to suppress SNS output to BAT. Thus, it is not clear whether TMX-mediated effect on BAT function is directly due to impaired estrogen signaling in BAT or due to reduced SNS output to BAT. Immunolabeling of tyrosine hydroxylase, a marker for sympathetic neurons, in male and female BAT may be considered.

Reviewer #2

(Remarks to the Author)

In this study, the authors investigate the role mechanism(s) of PGC-1 α in sexual dimorphism. Inducible adipocyte-specific PGC-1 α knockout (KO) mice displayed decreased BAT thermogenesis only in females. Expression of ChREBP β and downstream de novo lipogenesis (DNL) related genes were both reduced only in female KO mice. BAT-specific knockdown of ChREBP β reduced the DNL-related gene expression and BAT thermogenesis in female wild-type mice. Furthermore, PGC-1 α enhanced the sensitivity of female BAT estrogen signaling, thereby increasing ChREBP β and its downstream DNL-related gene expression. These findings suggest that PGC-1 α -ChREBP β mediated DNL plays a pivotal role in BAT thermogenesis in a sex-dependent manner.

Major points

- 1- The stage of the estrus cycle is critical for interpretation of the data obtained in female mice.

- 2- Is UCP1 expression more elevated in BAT from females than from males ?
- 3- In Figure 3 and in Figure 7, the expression of ChREBP α (RT-qPCR) as well as ChREBP protein content should be measured (actually in males and in females)
- 4- What are the functional consequences of the OVX-treatment on mitochondrial shape, /structure/function ?
- 5-FGF21 , a key actor of BAT function, known to be regulated in a sexual dimorphic manner should be measured in all the experiments presented.

Reviewer #3

(Remarks to the Author)

The manuscript by Takeuchi et al found a female-specific effect of BAT thermogenesis most likely depending on PGC-1 α -CHREBP β mediated de novo lipogenesis. Overall, this study is well designed and executed and the findings are quite interesting. The major concern for this study is that lack of functional study on the DNL with only gene expressions of some DNL related genes, such as Acss2, FASN etc. It remains unknown why DNL in BAT is important for thermogenesis. Although the authors attempted to show the lipidomic profiles in BAT, especially cardiolipins. However, tetra linoleoyl CL is the major functional cardiolipins, and it is NOT dependent on DNL, since linoleic acid is an essential acid and can only be taken up from dietary sources. It is recommended to show some data that DNL is in fact important in the proposed mechanism using isotope tracing experiments, such as ^3H labeled glucose, or ^3H H $_2\text{O}$. Below are some minor concerns:

1. Please indicate the weights/mass of BAT in different mice.
2. Figure 1d and 1f, Male KO mice also showed similar trends with female mice although it is not statistically significant. Is it because of the larger individual variation?
3. How to explain the difference of GTT in male and female mice in Fig.s1f
4. IN Figure 4A, please justify why they only focused on polar metabolites using CE-MS methods. IN the heatmap, how do they normalize the different metabolites?
5. In Figure 5e and 5f, why the patterns of CL(18:2) are different in these two figures?
6. In line 644, what does "10 uL brown adipose tissue samples" mean?

Version 1:

Reviewer comments:

Reviewer #1

(Remarks to the Author)

Reviewer #2

(Remarks to the Author)

No more comments

My queries were addressed, congratulations on the mice work !

Reviewer #3

(Remarks to the Author)

The authors satisfactorily addressed my concerns

Reviewer #1

It is not clear how PGC-1 α promotes ChREBP β -mediated DNL. Does PGC-1 α interact with and coactivate ChREBP β ?

We appreciate the insightful comments provided by Reviewer #1 and the editor, which have highlighted the critical importance of elucidating the role of PGC-1 α in ChREBP β -mediated DNL. We fully recognize the significance of addressing this question and have conducted additional analyses to characterize the potential interaction between PGC-1 α and ChREBP β . We showed that PGC-1 α KO reduced *Chrebp β* expression at the transcriptional level in female BAT. Although the direct regulation of *Chrebp β* transcription by PGC-1 α has not been reported, PGC-1 α is known to influence the transcription of target genes through various histone modifications (PMID: 19008463). Therefore, we performed ATAC-seq analysis to assess chromatin accessibility at the relevant regulatory regions. Our results showed that the female BAT exhibited greater chromatin accessibility near the transcription start site (TSS) of *Chrebp β* than the male BAT (Fig. 3g), suggesting a sex-specific regulatory mechanism. Notably, this sex difference in chromatin accessibility was abolished only in female PGC-1 α knockout (KO) mice. These results indicate that PGC-1 α in female BAT regulates ChREBP β expression at the transcriptional level through the modulation of chromatin accessibility in a female-specific manner, rather than through direct interaction. Although further examinations are required to clarify the mechanism by which PGC-1 α modulates chromatin accessibility near the TSS of *Chrebp β* only in female mice, we will be committed to addressing this important question in our future studies.

These results have been added to the Results section (Lines 171–189, Figure 3d–g).

PGC-1 β has been shown to coactivate ChREBP β and promote DNL gene expression. Does PGC-1 α deletion alter PGC-1 β expression?

To answer this question, we additionally examined *Pgc1 β* expression levels in BAT. No significant difference in *Pgc1 β* expression was observed between male and female BAT. Moreover, *Pgc1 β* expression levels were not affected by PGC-1 α knockout (Fig. S1g). These results indicate that PGC-1 α regulates expression of ChREBP β and DNL-related genes in a PGC-1 β -independent manner.

These results have been added to the Results section of the revised manuscript (Lines 104–105, Supplementary Figure 1g).

It is not clear whether AAV-shChrebp targets specifically Chrebp β without altering Chrebp α expression. What is the effect of ChREBP β knockdown on mitochondrial membrane structure and lipid profiles in male BAT?

We evaluated the knockdown (KD) efficiency of AAV-shChrebp on *Chrebpβ* and *Chrebpα* expression in the BAT of both sexes. Our results demonstrated that AAV-shChrebp administration almost completely suppressed *Chrebpβ* expression while having no significant effect on *Chrebpα* expression in both male and female BAT (Fig. 6a, Fig. S5a, Fig. S6a-b). These results indicate that AAV-shChrebp selectively targets *Chrebpβ* expression in the BAT of male and female mice.

These results have been added to the Results section of the revised manuscript (Lines 252–254: Figure 6a; Supplementary Figure 5a, and Lines 273–274: Supplementary Figures 6a-b).

The reviewer requested examination of the effects of ChREBPβ KD on mitochondrial membrane structure and lipid profiles in **male** BAT. Furthermore, it appears that investigating the effects of ChREBPβ KD on lipid profiles in **female** BAT is necessary to assess its effect on the mitochondrial membrane structure. Therefore, we additionally analyzed the lipid profile in the BAT of female ChREBPβ KD mice.

We first examined the effect of ChREBPβ KD on mitochondrial membrane structure and lipid profiles in male BAT. In male mice, ChREBPβ KD did not affect the mitochondrial membrane structure (Fig. S6c–e). Therefore, we focused on the levels of cardiolipin (CL) and ether-linked phosphatidylethanolamines (PEs), which are known to maintain mitochondrial morphology and function (PMID: 29034233, 37069167, 38129691). Notably, CL(18:2)₄ and ether-linked PEs were found to be reduced only in female Pgc-1α KO mice (Fig. 5f–g, Fig. S5e). Interestingly, these lipid levels were not reduced in the BAT of male ChREBPβ KD mice (Fig. S6f–g).

Next, we examined the effects of ChREBPβ KD on CL(18:2)₄ and ether-linked PE levels in female BAT. In female BAT, ChREBPβ KD reduced ether-linked PE levels but did not affect CL(18:2)₄ levels (Fig. 6f–g). These results indicate that a female-specific ChREBPβ-dependent mechanism may enhance ether-linked PE levels in BAT, which could play a critical role in maintaining mitochondrial morphology and function. In contrast, ChREBPβ did not appear to be involved in PGC-1α-mediated CL(18:2)₄ production in female BAT.

In addition, we observed that PGC-1α KO did not affect *Chrebpα* expression in the BAT of either sex (Fig. S3a). Overall, these findings indicate that PGC-1α may regulate mitochondrial membrane structure through ChREBPβ-dependent ether-linked PE production and ChREBPβ-independent CL(18:2)₄ production in a female-specific manner. These results have been added to the Results section of the revised manuscript (Lines 273–279: Supplementary Figure 6a–g, and Lines 263–266: Figure 6f–g) and discussed in the Discussion section of the revised manuscript (Lines 370–373 and Lines 379–386).

Estrogen has been shown to enhance BAT function by increasing sympathetic nervous system (SNS) output to BAT (PMC4082097). In addition, mild cold (room temp)-induced AKT2 signaling has been shown to stimulate ChREBP-mediated DNL (PMC5762420). These findings indicate that increased SNS output to BAT in female mice can stimulate PGC-1 α expression and ChREBP-mediated DNL.

Estrogen receptor antagonist TMX is likely to suppress SNS output to BAT. Thus, it is not clear whether TMX-mediated effect on BAT function is directly due to impaired estrogen signaling in BAT or due to reduced SNS output to BAT. Immunolabeling of tyrosine hydroxylase, a marker for sympathetic neurons, in male and female BAT may be considered.

We appreciate the reviewer's insightful comment regarding the potential role of estrogen signaling and SNS output in regulating BAT function. To address this concern, we performed the immunolabeling of tyrosine hydroxylase (TH) and examined the mRNA levels of TH in the BAT of male and female wild-type mice treated with either TMX or vehicle for 5 days. Our results showed no significant differences in TH expression between the sexes or between the TMX and vehicle groups (Fig. S8a–b). These findings suggest that TMX-induced estrogen antagonism does not significantly affect SNS output to BAT in females. Furthermore, as shown in Figure 7g, TMX administration did not alter BAT *Pgc1 α* gene expression, further supporting the conclusion that TMX does not influence SNS output. These results have been added to the Results section of the revised manuscript (Lines 307–312: Supplementary Figure 8a–b).

Reviewer #2

The stage of the estrus cycle is critical for Interpretation of the data obtained in female mice.

As the reviewer pointed out, the estrus cycle might influence the interpretation of the experimental results. However, in our study, consistent phenotypes—including suppressed thermogenesis and mitochondrial structural deficits in female BAT, induced by either PGC-1 α KO or TMX treatment—were observed across all experiments conducted on **a single experimental day** (Fig. 1–7), without any special consideration of the estrus cycle.

To determine why this consistency was achieved, we collected blood samples from wild-type female mice during **a single experimental day**, as in our previous experiments, and measured serum estrogen levels. As shown in the figure below, we observed variability in serum estrogen levels among individual mice, which likely reflects the different stages of the estrus cycle on a single experimental day. Importantly, all experiments (Fig. 1–7) were performed under these conditions, where female mice had varying serum estrogen levels.

Despite this variability, consistent phenotypes were obtained, suggesting that the observed effects are not dependent on the estrus cycle stage. Furthermore, serum estrogen levels in male mice were found to be below the detection limit, consistent with a previous report (PMID: 25856427). These findings collectively suggest that even the lowest levels of estrogen in female mice are sufficient to account for the observed sex differences in BAT thermogenesis. Therefore, the estrus cycle stage may not be critical for interpreting the results obtained from female mice.

Is UCP1 expression more elevated in BAT from females than from males ?

We measured the expression levels of *Ucp1* in BAT. The expression levels of *Ucp1* in BAT were higher in females than in males (Fig. S1a), consistent with the enhancement of BAT thermogenesis in female mice.

These results have been added to the Results section of the revised manuscript (Lines 88–89: Supplementary Figure 1a).

In Figure 3 and in Figure 7, the expression of ChREBP α (RT-qPCR) as well as ChREBP protein content should be measured (actually in males and in females)

We measured *Chrebp α* gene expression levels and ChREBP protein levels in the BAT of both PGC-1 α KO mice and TMX-, an estrogen receptor antagonist, administered mice. Neither PGC-1 α KO (Fig. S3a) nor TMX administration (Fig. S7b) resulted in any changes in the expression levels of *Chrebp α* in either sex. Moreover, the expression levels of the ChREBP protein in BAT were higher in females than in males. In females, both PGC-1 α KO (Fig. S3b) and TMX administration (Fig. S7c) reduced BAT ChREBP protein expression levels, whereas in males, neither PGC-1 α KO nor TMX treatment induced such reductions.

These results have been added to the Results section of the revised manuscript (Lines 162–166: Supplementary Figure 3a–b, and Lines 294–296: Supplementary Figure 7b–c).

What are the functional consequences of the OVX-treatment on mitochondrial shape,

/structure/function ?

In the BAT of ovariectomized mice, the mitochondrial area was smaller, and the total length of the cristae was also reduced compared with that in the sham-operated female mice (Fig. S7g–i). In addition, VO_2 after NE administration was reduced in ovariectomized mice compared with that in sham-operated female mice (Fig. S7f). These results were similar to those observed in the TMX-administered female mice in this study. Overall, these results suggest that estrogen signaling plays an important role in BAT thermogenesis in female mice.

These results have been added to the Results section of the revised manuscript (Lines 302–306: Supplementary Figure 7f–i).

FGF21, a key actor of BAT function, known to be regulated in a sexual dimorphic manner should be measured in all the experiments presented.

We measured the expression levels of *Fgf21* in BAT in all experiments. No significant differences were observed in the expression levels of *Fgf21* between the control and intervention groups as well as between males and females (Fig. S3c, and as shown in the figure below). These results suggest that FGF21 did not play an important role in the sex difference in BAT thermogenesis in this model.

These results have been added to the Results section of the revised manuscript (Lines 166–168: Supplementary Figure 3c).

Reviewer #3

However, tetra linoleoyl CL is the major functional cardiolipins, and it is NOT dependent on DNL, since linoleic acid is an essential acid and can only be taken up from dietary sources. It is recommended to show some data that DNL is in fact important in the proposed mechanism using isotope tracing experiments, such as 3H labeled glucose, or 3H H_2O .

We deeply appreciate your valuable advice and suggestions. As suggested, we performed

lipidomics after D₂O administration and found that the labeled components of CL(18:2)₄ were not detected in the PGC-1α KO and Control mice of both sexes.. This result clearly indicates that PGC-1α does not enhance CL(18:2)₄ production through DNL in the BAT of female mice. In addition, we found that ChREBPβ knockdown did not affect CL(18:2)₄ levels in female BAT (Fig. 6f). Overall, these results suggest that PGC-1α increases CL(18:2)₄ levels through molecular mechanism(s) other than ChREBPβ-mediated DNL in female BAT.

In addition, we found that several molecular species of ether-linked phosphatidylethanolamine (PE), which were decreased in the female BAT of PGC-1α KO mice (Fig. S5e), were commonly reduced following BAT ChREBPβ knockdown in female mice (Fig. 6g). The labeled components of ether-linked PE(O-16:1_18:1) showed a significant decrease in lipidomics after D₂O administration in female Pgc-1α KO mice (Fig. 6h). These results suggest that ether-linked PE is synthesized via PGC-1α-ChREBPβ-mediated DNL. Recent studies have highlighted that ether-linked phospholipids play a crucial role in maintaining the morphology and function of the mitochondria (PMID: 37069167, 38129691).

In summary, PGC-1α may play a pivotal role in the mitochondrial function of female BAT through both ChREBPβ-dependent *de novo* ether-linked PE production and ChREBPβ-independent CL(18:2)₄ production.

These observations have been added to the Results section of the revised manuscript (Lines 263-272: Figure 6f–h, Supplementary Figure 5e) and discussed in the Discussion section of the revised manuscript (Lines 370-373 and Lines 379-386).

Below are some minor concerns:

1. Please indicate the weights/mass of BAT in different mice.

We measured the BAT mass in PGC-1α KO or TMX-, an estrogen receptor antagonist, administered mice. As shown in the figure below, neither PGC-1α KO nor TMX administration resulted in any change in BAT weight in either sex.

2. Figure 1d and 1f, Male KO mice also showed similar trends with female mice although it is not statistically significant. Is it because of the larger individual variation?

We agree that body temperature during cold exposure tended to be lower in Male KO mice than in Male Control mice. However, as the reviewer pointed out, these differences were not statistically significant. Moreover, the effects of PGC-1 α KO on BAT, such as energy expenditure (Figure 1g), gene expression (Figure 3a–c), metabolite profile (Figure 4a–e), lipid profile (Figure 5a–i), and mitochondrial structure (Figure 2a–c) and activity (Figure 2d and Figure 4e), were all observed only in female BAT, which is completely consistent with the fact that cold tolerance was significantly reduced only in female mice (Figure 1d, f). Therefore, the lack of a significant reduction in cold tolerance in male mice may be due to sex differences in BAT function.

3. How to explain the difference of GTT in male and female mice in Fig.s1f

A previous study (PMID: 22645355) reported that male adipocyte-specific PGC-1 α KO mice were systemic glucose intolerant. Our results are compatible with those of this previous report. Given that male mice have poorer glucose tolerance than female mice, impaired glucose tolerance may be more pronounced in male mice than in female mice, although further studies are required to clarify the mechanisms underlying this sex difference.

4. IN Figure 4A, please justify why they only focused on polar metabolites using CE-MS methods. IN the heatmap, how do they normalize the different metabolites?

We focused only on polar metabolites because most molecules produced when the mitochondrial TCA cycle and electron transport chain are activated during BAT thermogenesis are water-soluble and easily ionized molecules. CE-MS methods were used because they are suitable for analyzing polar metabolites.

In the heatmap analysis, we used the following formula for normalization:

$$Z = (X - \mu) / \sigma$$

(Z: standardized value of each metabolite, X: Relative area of each metabolite, μ : mean of each metabolite, σ : SD of each metabolite)

5. In Figure 5e and 5f, why the patterns of CL(18:2) are different in these two figures?

Figure 5e shows the proportion of each CL molecular species, including CL(18:2)₄, relative to the total lipid content in BAT. In contrast, Figure 5f presents the proportion of the CL(18:2)₄ molecular species normalized to the total CL content in BAT. The proportions of CL among

all lipids are very small, and those of individual CL species are greatly influenced not only by changes in the absolute amount of individual CL species but also by changes in the absolute amount of total lipids excluding CL. Therefore, to evaluate the changes in CL(18:2)₄ within CL, we measured the proportion of CL(18:2)₄ among CL as previously reported (PMID: 16855048) and found that CL(18:2)₄ was actually reduced only in female knockout mice, as shown in Figure 5f.

6. In line 644, what does "10 uL brown adipose tissue samples" mean?

We thank the reviewer for this comment. This is a typographical error; the correct notation should be "1 mg brown adipose tissue." We have corrected this error in the text.